# Role of TMEM100 in mechanically insensitive nociceptor un-silencing

Timo A. Nees[1,2], Na Wang[3], Pavel Adamek [4], Nadja Zeitzschel [4], Clement Verkest[4], Carmen La Porta[1], Irina Schaefer[1], Julie Virnich [4], Selin Balkaya[4], Vincenzo Prato[1], Chiara Morelli[5], Valerie Begay[6], Young Jae Lee [7], Anke Tappe-Theodor [1], Gary R. Lewin [6], Paul A. Heppenstall[5], Francisco J. Taberner[1,8] & Stefan G. Lechner [1,4] ✉

Mechanically silent nociceptors are sensory afferents that are insensitive to noxious mechanical stimuli under normal conditions but become sensitized to such stimuli during inflammation. Using RNA-sequencing and quantitative RT-PCR we demonstrate that inflammation upregulates the expression of the transmembrane protein TMEM100 in silent nociceptors and electrophysiology revealed that over-expression of TMEM100 is required and sufficient to un-silence silent nociceptors in mice. Moreover, we show that mice lacking TMEM100 do not develop secondary mechanical hypersensitivity−i.e., pain hypersensitivity that spreads beyond the site of inflammation−during knee joint inflammation and that AAV-mediated overexpression of TMEM100 in articular afferents in the absence of inflammation is sufficient to induce mechanical hypersensitivity in remote skin regions without causing knee joint pain. Thus, our work identifies TMEM100 as a key regulator of silent nociceptor un-silencing and reveals a physiological role for this hitherto enigmatic afferent subclass in triggering spatially remote secondary mechanical hypersensitivity during inflammation.

Pain is an unpleasant and multifaceted sensation that can be stabbing, burning, throbbing or prickling. Likewise, pain hypersensitivity has many faces and can manifest as increased sensitivity to heat, cold or mechanical stimuli that commonly spreads far beyond the site of initial insult, a phenomenon known as secondary hyperalgesia or allodynia. Our ability to distinguish this plethora of painful sensations, relies on the functional diversity of primary sensory afferents, which detect painful stimuli and relay information about the intensity and quality of these stimuli to the central nervous system. Functionally distinct subclasses of sensory afferents have already been discovered several decades ago by classical neurophysiological studies[1]. More recently, single-cell RNA-sequencing studies have mapped transcriptional signatures to these functionally classified neurons[2,3] and revealed changes therein associated with chronic pain[4,5]. Moreover, knock-out, cell ablation and optogenetic studies have deciphered the contribution of various afferent subclasses to different forms of pain, such as acute pain evoked by pinprick, pinch and punctate mechanical stimuli as well as mechanical allodynia and cold allodynia associated with nerve injury[6–10]. The sensory afferents that mediate cold allodynia are interesting, because they are normally 'silent' and only become sensitive to cold after nerve injury[10].

Another relatively large and still enigmatic population of silent nociceptors are the mechanically insensitive afferents (MIA), which only become sensitized to mechanical stimuli during inflammation.

[1]Institute of Pharmacology, Heidelberg University, Heidelberg, Germany. [2]Department for Orthopeadics, Heidelberg University Hospital, Heidelberg, Germany. [3]Institute of Pathophysiology, Yan'an University, Yan'an, China. [4]Department of Anesthesiology, University Medical Center Hamburg-Eppendorf, Hamburg, Germany. [5]SISSA: Scuola Internazionale Superiore di Studi Avanzati, Trieste, Italy. [6]Department of Neuroscience, Max Delbrück Center for Molecular Medicine, Berlin, Germany. [7]Department of Biochemistry, Lee Gil Ya Cancer and Diabetes Institute, Gachon University College of Medicine, Incheon, Republic of Korea. [8]Instituto de Neurociencias de Alicante, Universidad Miguel Hernández – CSIC, Alicante, Spain. ✉e-mail: s.lechner@uke.de

MIAs were first documented in the articular nerves of the cat knee joint[11] and were subsequently found in the colon[12,13] and the bladder[14] as well as in human skin[15]. It is estimated that MIAs constitute ~30% of all C-fiber afferents in viscera and joints and about 15–20% in the human skin, whereas they appear to be less abundant in mouse skin[16,17]. Considering the large proportion of MIAs in viscera and joints, it has been hypothesized that the un-silencing of MIAs during inflammation substantially increases nociceptive input onto the spinal cord, which supposedly potentiates central pain processing and eventually results in increased pain sensitivity. Microneurography from cutaneous human afferents, on the other hand, suggested that silent nociceptors might induce secondary mechanical hyperalgesia[18]. Owing to the lack of tools that would allow the unequivocal identification or the selective genetic manipulation of MIAs, neither the mechanism underlying the un-silencing of MIAs nor their exact role in pain signaling have hitherto been deciphered.

We had previously shown that visceral and deep somatic mouse MIAs express the nicotinic acetylcholine receptor alpha-3 subunit (CHRNA3) and can thus readily be identified in Tg(Chrna3-EGFP) BZ135Gsat reporter mice[19]. Most importantly, we had shown that CHRNA3-EGFP+ MIAs acquire mechanosensitivity upon treatment with the inflammatory mediator nerve growth factor (NGF), which also slightly increases the mechanosensitivity of polymodal nociceptors[20], and demonstrated that this process requires de-novo gene transcription. We thus here set out to identify transcriptional changes that underlie the un-silencing of CHRNA3-EGFP+ MIAs and to eventually utilize these findings to examine the contribution of MIAs to the generation of inflammatory pain.

## Results

### NGF treatment selectively upregulates TMEM100 in silent nociceptors

We had previously shown that cultured CHRNA3-EGFP+ MIAs acquire mechanotransduction currents after 24 h treatment with NGF[19]. To identify proteins required for this NGF-induced acquisition of mechanosensitivity we here compared the transcriptomes of CHRNA3-EGFP+ neurons, cultured in the absence or presence of NGF (50 ng/ml) for 24 h. To this end we manually picked CHRNA3-EGFP+ neurons with a patch-pipette and analyzed their transcriptomes using paired-end RNAseq (Fig. 1a). This comparison showed that neither the mechanically-gated ion channel *Piezo2*, which is required for mechanotransduction in CHRNA3-EGFP+ neurons[19], nor any of the known PIEZO2 modulators, such as *Stoml3, Pcnt, Mtmr2, Tmem150c, Cdh1, Anxa6, Atp2a2* and *Nedd4-2*[21–28], are up-regulated by NGF (Fig. 1b). Moreover, the analysis showed that CHRNA3-EGFP+ MIAs have a transcriptional signature−i.e., co-expression of *Ntrk1, Calca, Tac1, Trpv1, Nos1, Ly6e* and *Htr3a* but not *Cyp2j12, Prrx2* and *Etv1* (Fig. 1c)−that was previously observed in a subset of peptidergic nociceptors that were classified as PSPEP2 neurons in a large scale single-cell RNA-sequencing study[2]. However, the RNAseq screen revealed the NGF-induced up-regulation of the transmembrane protein TMEM100 (fold-change=3.805, *P* = 4.12E-5, *N* = 3 samples per condition, Fig. 1b), which attracted our attention because TMEM100 was previously shown to potentiate the activity of TRPA1[29]−an ion channel that plays an important role in pain signaling[30]−by releasing it from the inhibition by TRPV1 and both channels (TRPA1 and TRPV1) are expressed at significant levels in CHRNA3-EGFP+ neurons (Fig. 1c, d). Importantly, quantitative real-time PCR (qPCR) confirmed the NGF-induced up-regulation of TMEM100 in cultured CHRNA3-EGFP+ MIAs and further showed that no other major nociceptor sub-population exhibits significant changes in TMEM100 expression upon NGF treatment (Fig. 1e).

We thus next asked if the up-regulation of TMEM100 is involved in the acquisition of mechanosensitivity in MIAs. To this end we compared mechanotransduction currents of un-transfected CHRNA3-EGFP

+ neurons with currents form CHRNA3-EGFP+ cells that were transfected with a plasmid encoding TMEM100-IRES-dsRed-express2 using an electrophysiological approach known as the mechano-clamp technique[31]. Here, transmembrane currents are recorded from cultured DRG neurons in the whole-cell configuration of the patch-clamp technique while the cell soma is mechanically stimulated with a fire-polished patch-pipette. Consistent with our previous results[19], only a small proportion of un-transfected CHRNA3-EGFP+ cells (3/14) responded to mechanical stimulation with small inward currents (Fig. 1f–h). Strikingly, however, ~61% of the CHRNA3-EGFP+ neurons transfected with TMEM100 exhibited robust mechanotransduction currents that were significantly bigger than the small currents occasionally observed in control cells (Fig. 1g, h). When expressed in HEK293 cells, TMEM100 did not produce mechanotransduction currents nor did it modulate PIEZO2 mediated currents in these cells (Supplementary Fig. 1), indicating that TMEM100 is neither a channel itself nor a modulator of PIEZO2, but solely un-silences PIEZO2 in the specific cellular context of CHRNA3-EGFP+ MIAs.

### Intraarticular CFA injection induces knee joint pain and secondary allodynia in remote skin regions

To corroborate our in-vitro observations, we next examined the role of TMEM100 in the sensitization of MIAs in an in-vivo mouse model of Complete Freund's adjuvant (CFA)-induced knee joint monoarthritis. We chose knee joint inflammation as the experimental model because (i) MIAs were shown to constitute ~50% of all articular nociceptive afferents[11,19], (ii) because the levels of NGF, which induces upregulation of TMEM100, are significantly increased in synovial fluid in rodent models of inflammatory knee joint pain as well as in patients with osteoarthritis[32,33] and (iii) because anti-NGF antibodies alleviate joint pain in patients with osteoarthritis[34], suggesting that NGF and possibly MIAs, may play an important role in the generation of knee joint pain.

Consistent with our previous study[19], we found that the knee joint is densely innervated by CHRNA3-EGFP+ afferents that co-express calcitonin gene-related peptide (CGRP), which mostly terminate in Hoffa's fat pad (Fig. 2a–c). As previously described[35], intraarticular CFA injection caused prominent knee joint inflammation characterized by redness and swelling (Fig. 2d), which was accompanied by severe limping (Supplementary Movies S1 and S2)−indicative of primary hyperalgesia in the knee – and by secondary mechanical and thermal hypersensitivity in skin regions remote from the knee joint. Primary knee joint hyperalgesia was quantified with the Catwalk XT gait analysis system (Fig. 2e), which revealed that mice with an inflamed knee joint put less weight on the affected leg, evidenced by a reduction of the ratio of the foot print area of the ipsi- (left) and contralateral (right) hind paw (before, $1.07 \pm 0.03$ vs. 3 days post injection (dpi) CFA, $0.57 \pm 0.07$, $N = 16$, Students paired t-test, $P = 2.6 \times 10^{-7}$), and a reduction of the leg swing speed of the inflamed leg (before, $1.041 \pm 0.017$ vs. 3 dpi CFA, $0.634 \pm 0.044$, $N = 16$, Students paired t-test, $P = 8 \times 10^{-9}$; Fig. 2f). Since the Catwalk XT assay might not reveal all aspects of knee joint pain, we also monitored other behaviors that might be indicative of pain using the LABORAS system (laboratory animal behavior observation registration and analysis system, METRIS b.v) for an observation period of 16 h before and after saline and CFA injection, respectively. We did, however, not detect any significant changes in the frequency of episodes of grooming, drinking or immobility, which might be indicative of severe pain, but only observed a reduction in the total distance traveled and the frequency of rearing (Supplementary Fig. 2). In addition to primary knee joint pain (Catwalk XT) and overall well-being (LABORAS), we also assessed secondary mechanical and thermal hypersensitivity in the ipsilateral hind paw using the von Frey and Hargreaves tests, respectively. These tests showed that CFA-induced monoarthritis, markedly reduces the minimal force of punctate mechanical stimuli−applied with von Frey filaments to the plantar

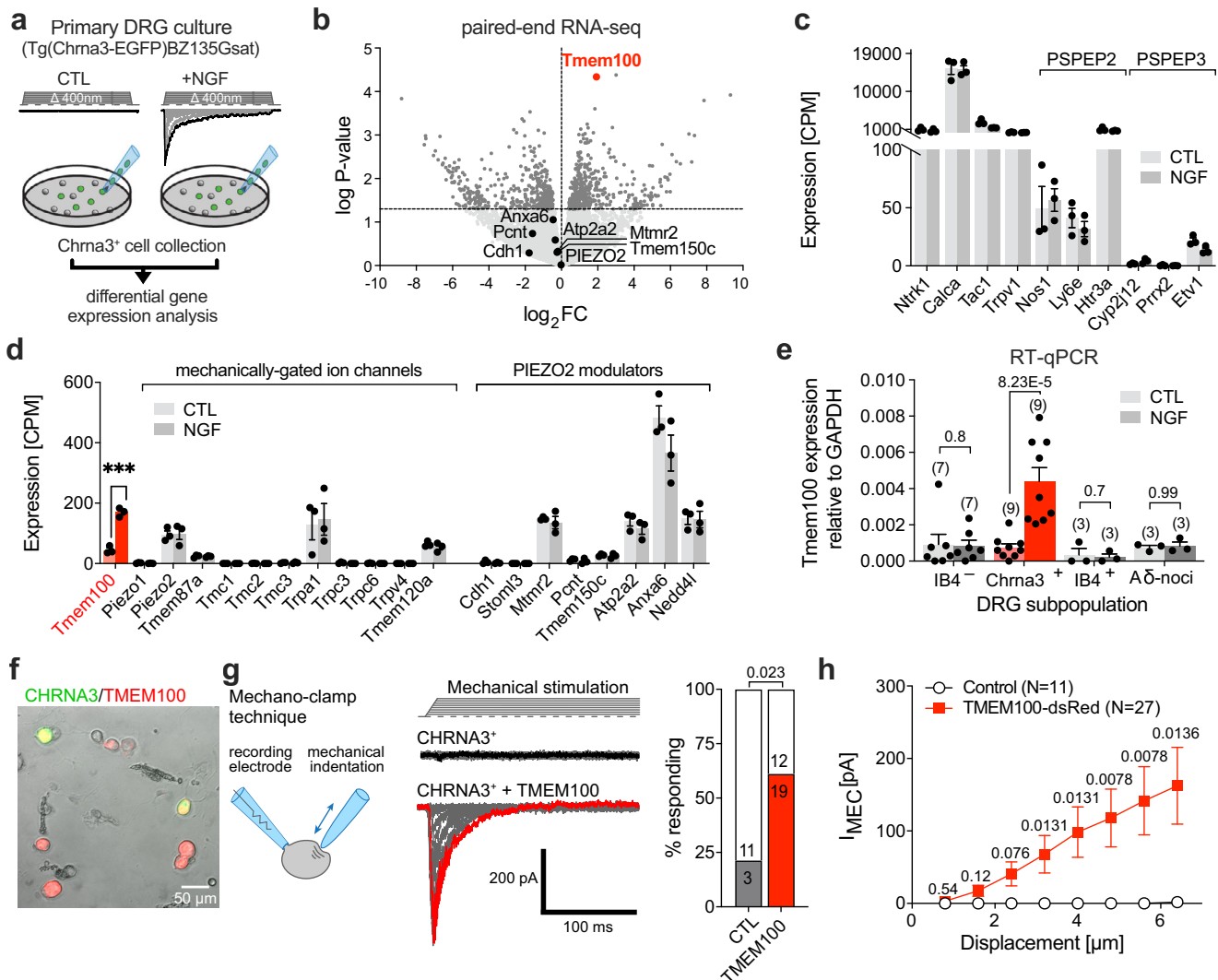

**Fig. 1 | NGF treatment selectively upregulates TMEM100 in silent nociceptors.**
**a** Cartoon depicting the RNAseq screen workflow. **b** Volcano plot showing the mean fold-change of expression upon 24 h NGF treatment ($\log_2$FC) vs. the log $P$-value determined by paired-end RNAseq ($n$ = 3 biologically independent samples, 20 cells per sample, from three different mice; two-sided Student's T-test). **c** Comparison of the mean ± SEM expression levels (counts per million, CPM) of peptidergic nociceptor subclass markers and **d** mechanically-gated ion channels and PIEZO2 modulators, determined by RNAseq in CHRNA3-EGFP⁺ neurons cultured with and without NGF, using two-sided Students t-test (***, $P$ = 4.59843E-05; $N$-numbers are the same as in **b**). **e** Comparison of the mean ± SEM expression levels of TMEM100 (normalized to GAPDH expression) determined by qPCR in the indicated nociceptor subclasses, cultured without (CTL) and with NGF. To enable identification of peptidergic C-fiber nociceptors, non-peptidergic C-fiber nociceptors and Aδ-fiber nociceptors for sample collection, cultures were prepared from Tg(Npy2r-cre) SM19Gsat/Mmucd x B6;129S-Gt(ROSA)26Sortm32(CAG-COP4*H134R/EYFP)Hze/J (Npy2rᶜʳᵉ;ChR2-EYFP) mice, in which Aδ-fiber nociceptors express EYFP and were additionally labeled with Alex-Fluor-568 conjugated Isolectin B4 (IB4), which

selectively binds to non-peptidergic C-fiber nociceptors. Numbers of samples (20 cells each) per subpopulation are indicated in brackets above the bars and individual values are shown as dots (two-sided Mann-Whitney test: $P$-values provided above bars). **f** Image showing cultured DRG neurons from CHRNA3-EGFP mice transfected with TMEM100-dsRed. **g** Cartoon depicting the mechano-clamp configuration of the patch-clamp technique (left, originally published in Verkest et al. https://doi.org/10.1038/s41467-022-28974-6 Nat Commun), example traces of mechanically-evoked currents in CHRNA3-EGFP⁺ control cells (middle, top) and in TMEM100-dsRed-transfected CHRNA3-EGFP⁺ cells (middle, bottom) as well as bar graph showing the proportion of cells responding to mechanical stimulation. Proportions were compared with two-sided Fisher's exact test ($P$ = 0.023). **h** Mean ± SEM peak amplitudes of mechanically-evoked currents are shown as a function of membrane displacement for control (white circles) and TMEM100 transfected cells (red squares). Current amplitudes were compared using multiple two-sided Mann-Whitney tests ($P$-values are provided above the symbols). N-numbers differ from **g**, because some recordings crashed before maximal mechanical stimulation. Source data are provided as a Source Data file.

surface of the hind paw—required to evoked a paw withdrawal reflex from 0.87 ± 0.03 g before to 0.15 ± 0.03 g three days post CFA injection (Student's paired t-test, $P$ = 5.18 × 10⁻¹²; Fig. 2g). Likewise, the latencies of paw withdrawals evoked by heat stimulation of the hind paw were also significantly reduced (before, 6.03 ± 0.28 s vs. 3 dpi CFA 2.31 ± 0.11 s $N$ = 16, Student's paired t-test, P = 1.8 × 10⁻⁹, Fig. 2h). It should be noted that it is a matter of intense debate whether reflexive paw withdrawal evoked by stimulation with von Frey filaments indicates pain or solely mechanical hypersensitivity of nociceptors[36] and

thus we will hereafter refer to a reduction in paw withdrawal thresholds as secondary mechanical hypersensitivity.

## CFA-induced knee joint inflammation induces mechanosensitivity and potentiates TRPA1 activity in CHRNA3-EGFP⁺ afferents

To enable the examination of CFA-induced transcriptional and functional changes in MIAs, we next back-labeled sensory neurons that give rise to articular afferents by intraarticular injection of the retrograde

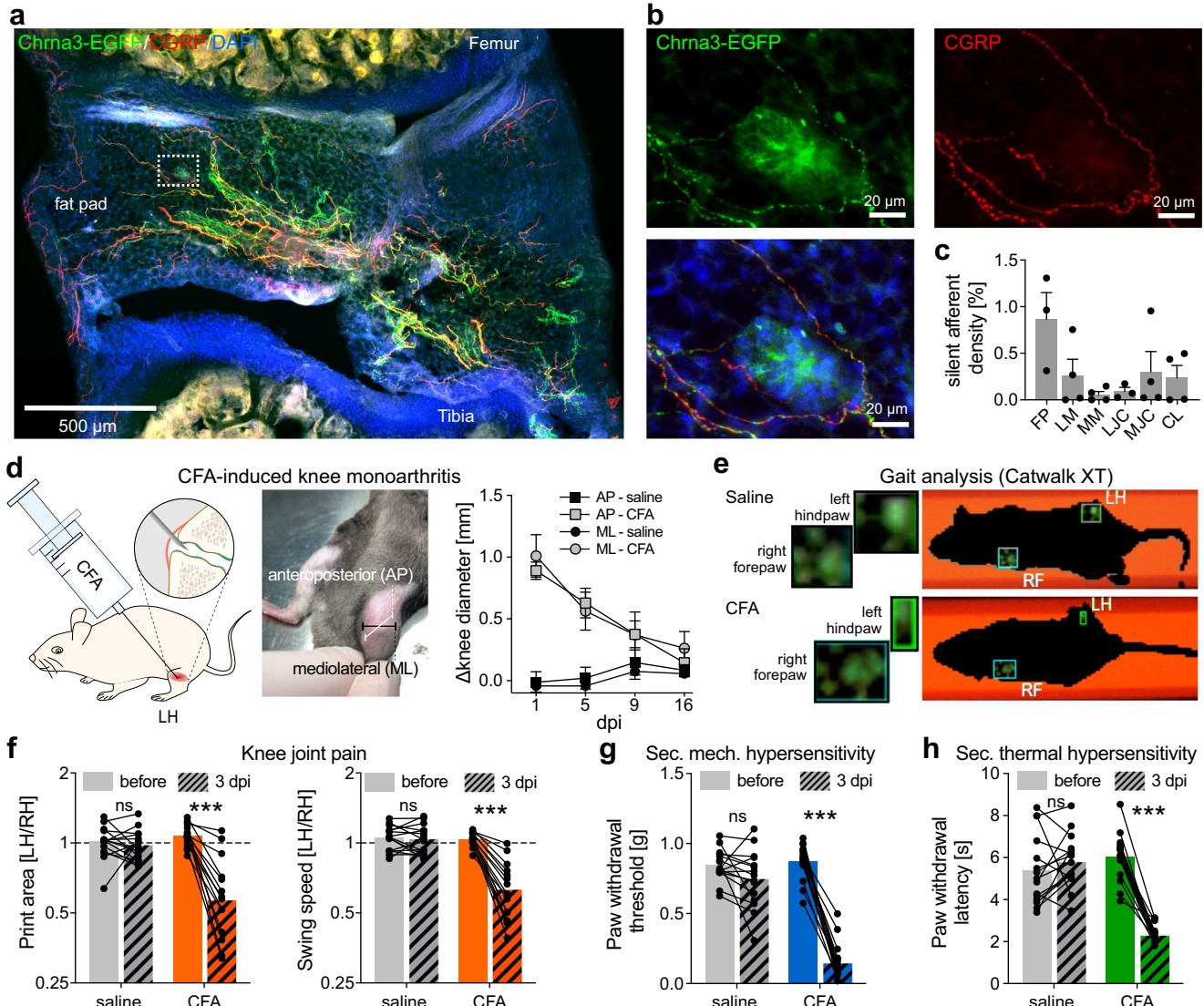

**Fig. 2 | Intraarticular CFA injection induces knee joint pain and secondary mechanical hypersensitivity in the hind paw. a** Representative image of a knee joint section immunostained for CGRP and EGFP. Note, only EGFP⁺ fibers that co-express CGRP are sensory afferents. CGRP⁻/EGFP⁺ fibers are sympathetic efferents (see Prato et al., 2017). **b** Close-up of the region marked by the white rectangle in **a**, emphasizing co-expression of EGFP and CGRP. **c** Quantification of silent afferent density (EGFP⁺/CGRP⁺ fibers) in anatomically defined regions. FP (Hoffa's fat pad), LM (lateral meniscus), MM (medial meniscus), LJC (lateral joint capsule), MJC (medial joint capsule), CL (cruciate ligament). Bars represent means ± SEM from three different mice. For each mouse, at least three photomicrographs per anatomical region were analyzed and averaged. Means from each mouse are shown as black dots. **d** Cartoon depicting the experimental approach (left), photograph of an inflamed knee (middle) and time course (days post-injection, dpi) of saline- (*n* = 5 mice) and CFA-induced (*n* = 6 mice) changes in knee size (right). Data are presented as means ± SEM. **e** Freeze frames of Catwalk XT movies from saline (top) and CFA (bottom) treated mice. Note, CFA-treated mice only put little weight on the left hind paw (small foot print). **f** Comparison of the foot print area ratio (left/right hindpaw; LH/RH) and the leg swing time ratio (LH/RH) measured before (solid bars) and three day after (3 dpi, hatched bars) saline (gray) and CFA (orange) injection. Two-sided paired Student's t-test (saline *N* = 15, CFA *N* = 16; print area CFA, *P* = 2.6 × 10⁻⁷; swing speed CFA P = 8 × 10⁻⁹). **g** Comparison of mechanical paw withdrawal thresholds before (solid bars) and three day after (3 dpi, hatched bars) saline (gray) and CFA (blue) injection. Paired two-sided Student's t-test (saline *N* = 15, CFA *N* = 16; CFA, *P* = 5.18 × 10⁻¹²). **h** Comparison of thermal paw withdrawal latencies before (solid bars) and three day after (3 dpi, hatched bars) saline (gray) and CFA (blue) injection. Paired two-sided Student's t-test (saline *N* = 15, CFA *N* = 16; CFA, *P* = 1.8 × 10⁻⁹). Source data are provided as a Source Data file.

tracer Fast Blue (FB) (Fig. 3a). Quantification of FB⁺ cells in serial sections of L3 and L4 DRGs, showed that this approach labeled a total of ~340 DRG neurons (191.5 ± 39.8 cells in L3 DRGs and 150.3 ± 56.7 cells in L4 DRGs, Fig. 3a). IB4-labeling of DRG cultures from FB-injected mice, further showed that 35.1% (108/308 FB⁺ cells) of the FB⁺ cells were CHRNA3-EGFP⁺, 25.3 % (78/308 FB⁺ cells) were small diameter (<30 μm) IB4⁻ peptidergic nociceptors and 26 % (80/308 FB⁺ cells) were large diameter neurons (Fig. 3b) that most likely give rise to group II articular afferents that detect innocuous stimuli. Only a small proportion of the retrogradely labeled neurons were IB4⁺ (13.6%, Fig. 3b), demonstrating

that the great majority of nociceptive knee joint afferents are peptidergic (IB4⁻ and CHRNA3-EGFP⁺) and that CHRNA3-EGFP⁺ neurons account for ~47 % (108/228 FB⁺ nociceptors) of all articular nociceptive afferents. To test if intra-articular CFA injection up-regulates TMEM100 expression in knee joint afferents, we manually collected small diameter (<30 μm) IB4⁻/FB⁺ and CHRNA3-EGFP⁺/FB⁺ cells from acutely dissociated ipsi- and contralateral L3 and L4 DRGs (3 h in culture) 3 days after CFA treatment and measured TMEM100 expression levels using qPCR. This analysis revealed that, similar to in-vitro NGF treatment, CFA-induced knee joint inflammation selectively

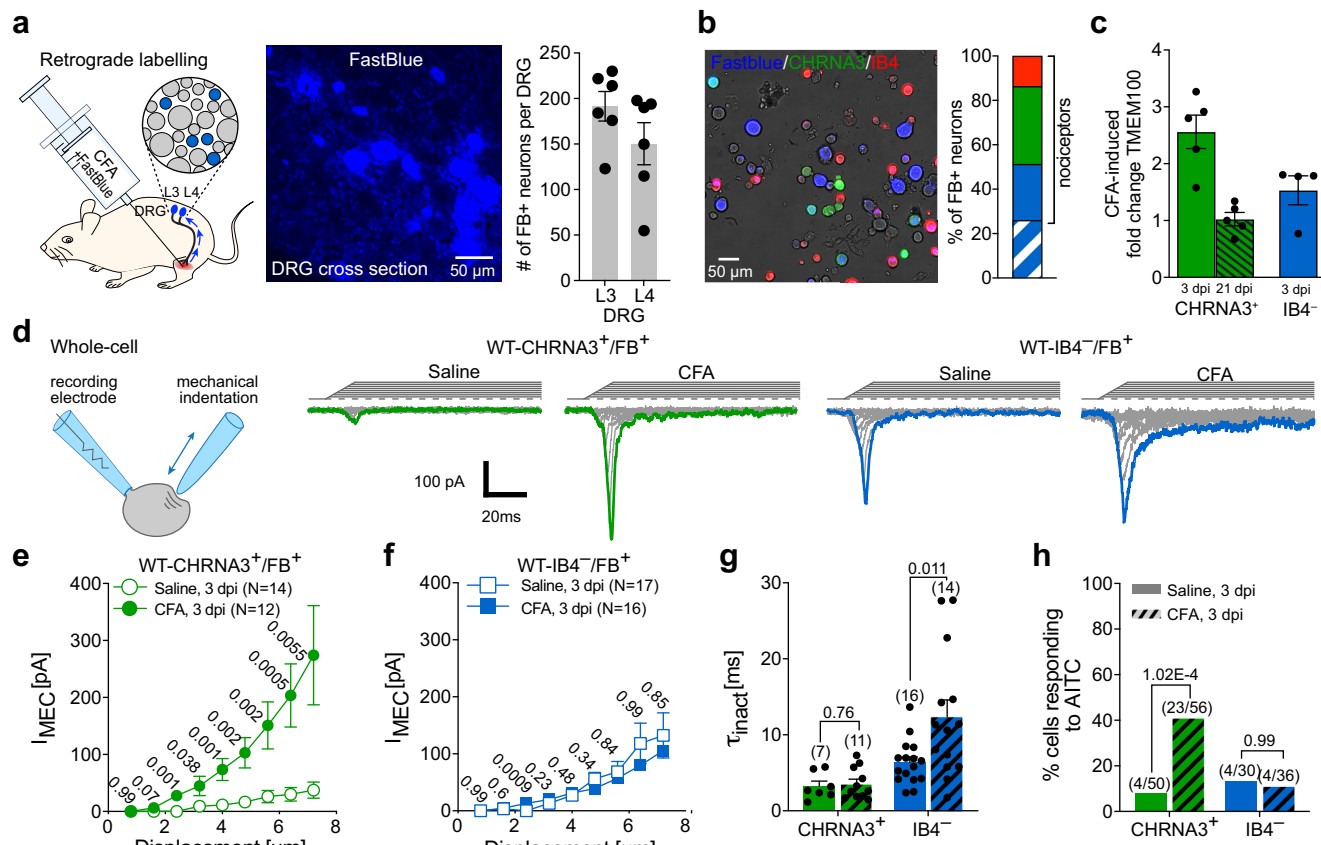

**Fig. 3 | CFA-induced knee joint inflammation induces mechanosensitivity and potentiates TRPA1 activity in CHRNA3-EGFP+ afferents. a** Cartoon depicting retrograde labeling of knee joint afferents (left), example image of FB+ neurons in DRG section (middle) and quantification of the mean ± SEM numbers of FB+ neurons in L3 and L4 DRGs from 6 mice. Dots represent individual values for each DRG. **b** Example image of a DRG culture from a CHRNA3-EGFP mouse after intraarticular FB (left). Stacked bar graph (right) shows the proportions of IB4+ cells (non-peptidergic nociceptors, red), MIAs (green), peptidergic nociceptors (IB4−, <30 μm, blue) and group-II articular afferents (IB4−, > 30 μm, blue hatched). **c** Quantification of CFA-induced changes in TMEM100 expression in FB-labeled MIAs and peptidergic nociceptors. Expression levels in ipsi- and contralateral DRGs (20 cells per sample), at the indicated timepoints, were compared by qPCR (ΔΔCt method). Bars represent means ± SEM and data from individual mice are shown as black dots (N = 5 mice for 3 and 21 dpi in CHRNA3+/FB+ cells and N = 4 for IB4−/FB+ cells). **d** Example traces of mechanically-evoked currents. **e**, **f** Comparison of the

mean ± SEM peak amplitudes of mechanotransduction currents evoked by increasing membrane displacements of **e** MIAs and **f** peptidergic nociceptors from saline (open symbols) and CFA-treated (solid symbols) mice, using two-sided Mann-Whitney test (P-values are provided next to the symbols in **e** and **f**). **g** Comparison of the mean ± SEM inactivation time constants of the mechanically-evoked currents using two-sided T-test. P-values and N-numbers of independent experiments (numbers in brackets) are shown above the bars. **h** Comparison of the proportions of CHRNA3+/FB+ (green) and IB4−/FB+ (blue) DRG neurons from wild-type mice that exhibit Ca2+ transients (visualized with Calbryte-590, see Supplementary Fig. 3a, b) in response to the TRPA1 agonist AITC (10 μM) 3 dpi of saline (solid fill) or CFA (cross-hatched bars) using two-sided Fishers exact test (P-values above bars). Numbers above bars indicate the number of responders and number of tested cells. Panel d drawing originally published in Verkest et al. https://doi.org/10.1038/s41467-022-28974-6 Nat Commun Source data are provided as a Source Data file.

up-regulates TMEM100 in CHRNA3-EGFP+ MIAs but not in other C-fiber nociceptors (Fig. 3c). Interestingly TMEM100 expression levels only increased transiently and returned to baseline values 21 after CFA treatment (Fig. 3c).

We next asked if mechanosensitivity of FB-labeled DRG neurons is altered in CFA-induced monoarthritis using the afore-describe mechano-clamp technique. In accordance with our previous results[19], CHRNA3-EGFP+/FB+ neurons from saline injected animals did not show currents in response to mechanical stimulation of the plasma membrane (Fig. 3d, e). Following intraarticular CFA-injection (3 dpi), however, FB-labeled CHRNA3-EGFP+ neurons exhibited robust mechanotransduction currents that were significantly larger than the small inward currents occasionally observed in control animals (Fig. 3d, e). Interestingly, the amplitudes of the mechanotransduction currents of small-diameter IB4− nociceptors were not altered in CFA-treated mice (Fig. 3f), but we observed a small, yet significant, increase in the inactivation time constants of these currents (Fig. 3g). Since TMEM100 was previously shown to potentiate the activity of TRPA1[29],

we also examined the responsiveness of FB-labeled neurons to the TRPA1 agonist allylisothiocyanate (AITC) using Calbryte-590 Ca2+-imaging. Strikingly, the proportion of CHRNA3-EGFP+/FB+ knee joint MIAs that responded to AITC was markedly increased from 8% (4/50 cells) in saline-treated mice to 41% (23/56 cells) in CFA-treated mice (Fig. 3h). Interestingly the proportion of AITC sensitive small diameter IB4−/FB+ neurons (putative polymodal C-fiber nociceptors) was not altered by CFA treatment (Fig. 3h), which was consistent with our observation that TMEM100 expression is not altered in these cells (Fig. 3c). We also observed a notable increase in the average response amplitude of CHRNA3-EGFP+ neurons, but given the small number of responding cells from saline-treated mice, this effect is difficult to interpret and might not be very meaningful (Supplementary Fig. 3a, b).

Taken together, our data showed that intraarticular CFA-injection causes knee joint pain and secondary hyperalgesia in the ipsilateral hind paw, which is accompanied by an upregulation of TMEM100, the potentiation of TRPA1 activity and, most importantly, the acquisition of mechanosensitivity in CHRNA3-EGFP+ neurons.

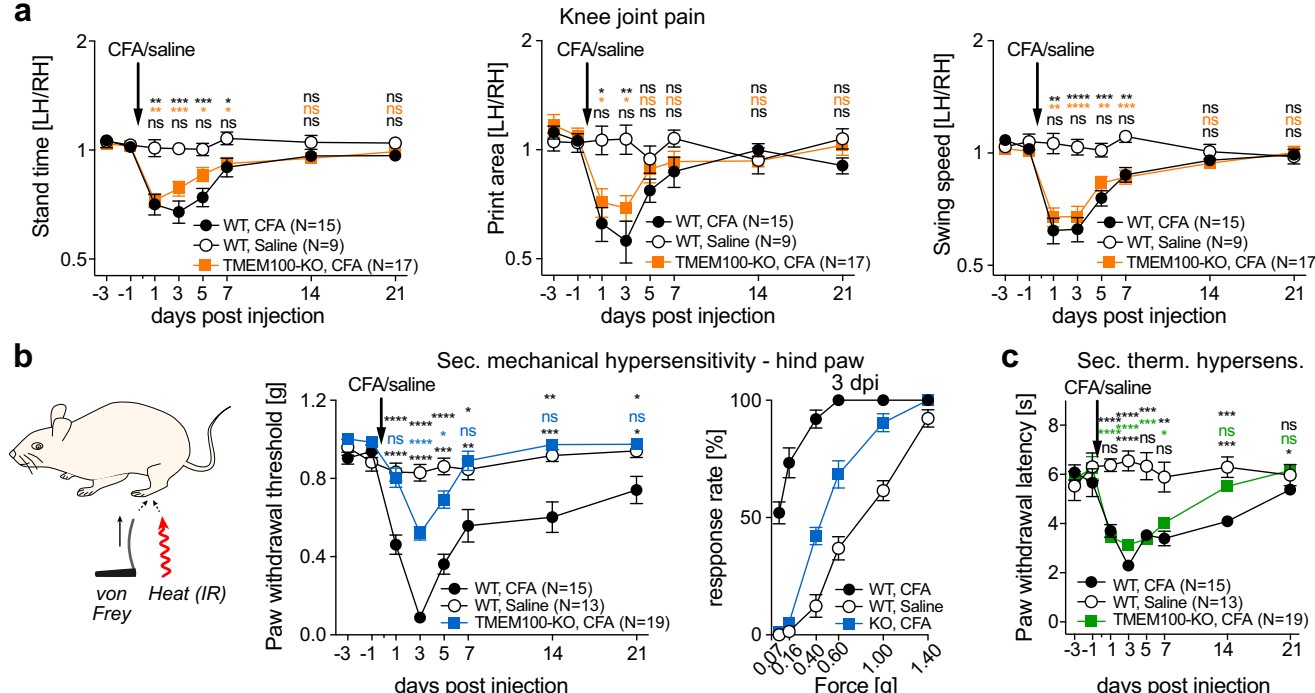

**Fig. 4 | TMEM100 knock-out mice develop normal inflammatory knee joint pain but no long-lasting secondary mechanical hypersensitivity. a** Comparison of the time courses of changes in stand time (left), foot print area (middle) and leg swing speed (right) of saline injected WT mice (white circles), CFA injected WT mice (black circles) and CFA injected TMEM100KO mice (orange squares). **b** Cartoon depicting the experimental approach for measuring secondary mechanical and thermal hypersensitivity in the ipsilateral hindpaw (left), time courses of changes in mechanical paw withdrawal thresholds of saline injected WT mice (white circles), CFA injected WT mice (black circles) and CFA injected TMEM100KO mice (blue squares) (middle) and responsiveness of WT mice (white circles), CFA injected WT mice (black circles) and CFA injected TMEM100KO mice (blue squares) three days post saline/CFA injection (3 dpi) to all tested von Frey filaments (right). Response rates were compared using 2-way ANOVA. *P*-values of multiple comparisons are as follows: WT-saline vs. WT-CFA, $P_{0.07g}$ = 7.5E-08, $P_{0.16g}$ = 2.8E-08, $P_{0.4g}$ = 8.7E-12,

$P_{0.6g}$ = 7.5E-08, $P_{1g}$ = 2.6E-06, $P_{1.4g}$ = 1.2E-01; WT-saline vs. TMEM100KO-CFA, $P_{0.07g}$ = 5.9E-01, $P_{0.16g}$ = 3.3E-01, $P_{0.4g}$ = 1.5E-04, $P_{0.6g}$ = 7.8E-04, $P_{1g}$ = 6.8E-05, $P_{1.4g}$ = 1.2E-01; WT-CFA vs. TMEM100KO-CFA, $P_{0.07g}$ = 5.2E-08, $P_{0.16g}$ = 3.6E-08, $P_{0.4g}$ = 3.9E-10, $P_{0.6g}$ = 9.9E-05, $P_{1g}$ = 6.1E-02, $P_{1.4g}$ = not determined). **c** Time courses of changes in thermal paw withdrawal latencies of saline injected WT mice (white circles), CFA injected WT mice (black circles) and CFA injected TMEM100KO mice (green squares). **a–c** Symbols represent means ± SEM. Unless otherwise stated, ratios at different time points were compared using mixed model ANOVA and *P*-values of multiple comparisons are indicated above the white circles. Top, WT-saline vs. WT-CFA; middle, WT-saline vs. TMEM100KO-CFA; bottom, WT-CFA vs. TMEM100KO-CFA (*$P$ < 0.05; **$P$ < 0.01; ***$P$ < 0.001; ns, $P$ > 0.05, N-numbers are provided in the graph legends). Exact *P*-values and detailed statistical information is provided together with the source data as a Source Data file.

## TMEM100 knock-out mice develop normal inflammatory knee joint pain but no long-lasting secondary mechanical allodynia

We next asked if the pain phenotype of CFA-induced knee monoarthritis (Fig. 2f–h) is causally linked to the sensitization of CHRNA3-EGFP[+] nociceptors and if this sensitization is induced by the upregulation of TMEM100. To this end we generated conditional TMEM100 knock-out mice, hereafter referred to as TMEM100KO mice, by crossing mice that carry a conditional allele for TMEM100[37] with SNS-Cre mice, in which Cre-recombinase expression is driven by the voltage-gated sodium channel Na$_v$1.8 promoter[38] and is thus expressed in all nociceptors including CHRNA3-EGFP[+] neurons[39]. We first compared primary knee joint pain, assessed by Catwalk XT gait analysis, in male wildtype (WT) mice that received intraarticular saline injections with male WT and TMEM100KO mice that received CFA injections, over a period of 21 days. WT mice that received CFA, exhibited significantly altered gait, indicative of knee joint pain, during the first seven days post CFA injection compared to saline treated animals. Surprisingly, CFA-treated TMEM100KO mice also developed knee joint pain and showed altered gait (Supplementary movie S3), which significantly differed from saline-treated WT mice (Supplementary movie S1) and was indistinguishable from gait observed in CFA-treated WT mice (Supplementary movie S2; Fig. 4a). Strikingly, however, secondary mechanical hypersensitivity in the ipsilateral hind paw was significantly attenuated in TMEM100KO mice. Thus, the von Frey paw

withdrawal thresholds were only transiently reduced from day 3 until day 5 and returned to baseline values by day 7 in TMEM100KO mice, while WT mice exhibited long-lasting secondary mechanical allodynia, which persisted until the end of the examination period (21 dpi, Fig. 4b). Secondary thermal hypersensitivity was not altered in TMEM100KO mice (Fig. 4c).

Since an increasing body of literature demonstrates sex differences with regards to pain sensitivity, we reproduced the behavioral experiments using female TMEM100KO mice. Interestingly, female mice exhibited the exact same pain phenotype as male mice with respect to primary and secondary hypersensitivity (Supplementary Fig. 4) indicating a sex-independent role of TMEM100 in inflammatory pain.

## CFA-induced knee joint inflammation fails to sensitize CHRNA3-EGFP[+] neurons to mechanical stimuli in TMEM100 knock-out mice

We next examined the role of TMEM100 in the acquisition of mechanosensitivity and the potentiation of TRPA1 activity in CHRNA3-EGFP[+] knee joint afferents in CFA-induced monoarthritis. Patch-clamp recordings from FB-labeled articular MIAs showed that CHRNA3-EGFP[+] neurons from TMEM100KO mice did not acquire mechanosensitivity during CFA-induced inflammation (Fig. 5a, b) and that mechanosensitivity of FB[+]/IB4[−] neurons was also not altered (Fig. 5a–d).

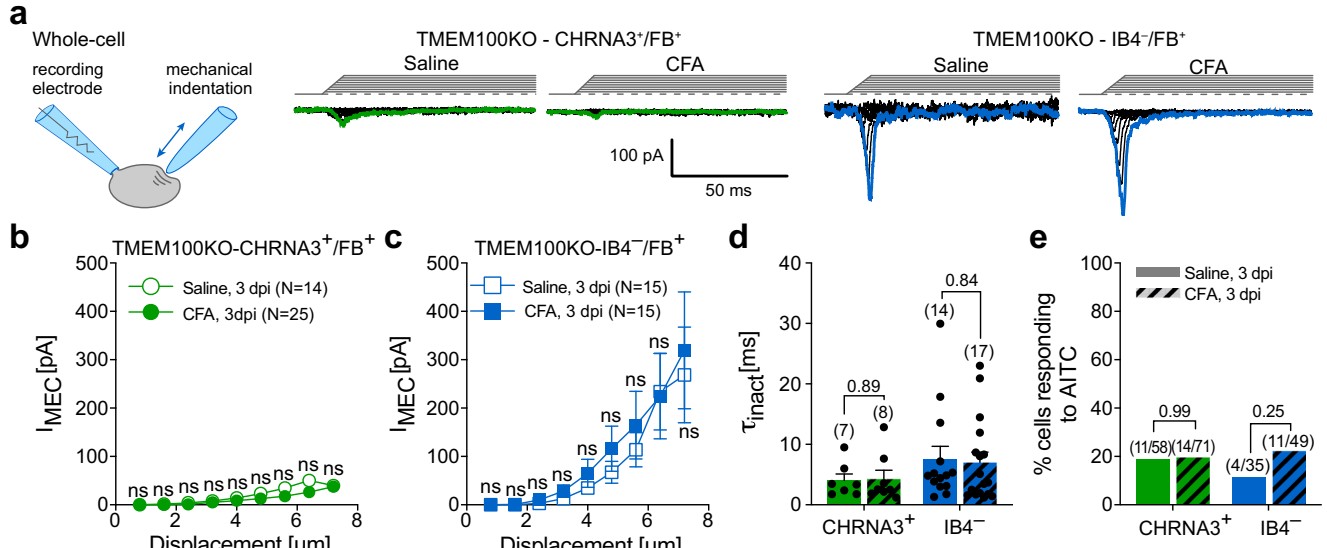

**Fig. 5 | CFA-induced knee joint inflammation fails to sensitize CHRNA3-EGFP⁺ neurons to mechanical stimuli in TMEM100 knock-out mice. a** Cartoon depicting the mechano-clamp configuration of the patch clamp technique (left, originally published in Verkest et al. https://doi.org/10.1038/s41467-022-28974-6 Nat Commun) and example traces of mechanically-evoked currents in the indicated cell types and conditions (right). Note, in contrast to WT mice, CHRNA3-EGFP⁺ neurons from TMEM100KO mice do not acquire mechanosensitivity during CFA-induced inflammation. **b, c** mean ± SEM peak amplitudes of mechanically-evoked currents are shown as a function of membrane displacement for **b** MIAs and **c** peptidergic nociceptors from saline (open symbols) and CFA-treated (solid symbols) mice. Current amplitudes were compared using two-sided Mann-Whitney test (P-values for **b** from left to right: ns = 0.99, ns = 0.45, ns = 0.21, ns = 0.57, ns =

0.33, ns = 0.21, ns = 0.22, ns = 0.12, ns = 0.55; P-values for **c** from left to right: ns = 0.99, ns = 0.99, ns = 0.25, ns = 0.4, ns = 0.73, ns = 0.99, ns = 0.47, ns = 0.5, ns = 0.92). **d** Comparison of the mean ± SEM inactivation time constants of mechanically evoked using two-sided unpaired t-tests (P-values are provided above bars and individual values are shown as dots. **e** Comparison of the proportions of CHRNA3⁺/FB⁺ (green bars) and IB4⁻/FB⁺ (blue bars) DRG neurons from TMEM100KO mice that exhibit Ca²⁺ transients (visualized with Calbryte-590, also see Supplementary Fig. 3c, d) in response to the TRPA1 agonist AITC (10 μM) 3 days post injection (3 dpi) of saline (solid fill) or CFA (cross-hatched bar) using two-sided Fishers exact test (P-values are indicated above the bars. Numbers in brackets above the bars indicate the number of responders and number of tested cells. Source data are provided as a Source Data file.

Moreover, in accordance with the previously proposed role of TMEM100 in disinhibiting TRPA1[29], neither the proportion of AITC-sensitive CHRNA3-EGFP⁺ knee joint afferents (Fig. 5e) nor the magnitude of the responses were altered by CFA treatment in TMEM100KO mice (Supplementary Fig. 3c). AITC responses of small diameter IB4⁻ nociceptors were also not altered (Fig. 5e and Supplementary Fig. 3d). Considering the small number of AITC-sensitive IB4⁻/FB⁺ cells from saline treated animals, definitive conclusions about differences in AITC response magnitude can, however, not be drawn.

Hence, our data indicates that CFA-treatment selectively upregulates TMEM100 in CHRNA3-EGFP⁺ MIAs (Fig. 3c), where it induces the acquisition of mechanosensitivity and the potentiation of TRPA1 (Figs. 3e–h and 5 and Supplementary Fig. 3). These findings together with the observation that paw withdrawal thresholds, but not knee joint pain, was greatly attenuated in TMEM100KO mice during CFA-induced knee inflammation (Fig. 4), suggest that the un-silencing of articular MIAs triggers the development of secondary mechanical hypersensitivity.

### Sensitization of cutaneous C-fiber nociceptors contributes to secondary mechanical allodynia

It is well established that central sensitization—i.e., a strengthening of synaptic transmission in pain processing circuits in the spinal cord—contributes to the development of secondary mechanical hypersensitivity[40,41]. Considering the remarkable reduction of paw withdrawal thresholds in CFA-induced knee joint monoarthritis (Fig. 4b), however, we asked if sensitization of cutaneous nociceptors also plays a role. To test this hypothesis, we directly measured the mechanosensitivity of C-fiber and Aδ-fiber nociceptors in the tibial nerve, which innervates the plantar surface of the hind paw, by recording mechanically evoked action potentials from single nerve

fibers in an ex-vivo skin-nerve preparation from mice that had received intraarticular CFA. In each single unit recording, we first determined the conduction velocity (CV) of the fiber for classification as a C-fiber (CV < 1 m/s) or an Aδ-fiber (CV 1–10 m/s) and the mechanical activation threshold using von Frey filaments, which were also used to determine the paw withdrawal thresholds in the behavioral experiments (Fig. 6a–c and Supplementary Fig. 5). The recordings revealed, that 40% of the cutaneous C-fiber nociceptors from mice with CFA-induced knee monoarthritis, are activated by von Frey filaments of 0.16 g and below and virtually all C-fibers (90%) responded to von Frey filaments of 0.4 g and below (Fig. 6c). By contrast, not a single C-fiber nociceptor from saline-treated control mice responded to von Frey stimuli ≤ 0.16 g and only ~17% were activated by the 0.4 g filament (Fig. 6c). Likewise, the proportion of cutaneous Aδ-fiber nociceptors that responded to von Frey stimuli ≤ 0.16 g was also significantly larger in CFA-treated WT mice, but this difference was less pronounced than in C-fibers (Supplementary Fig. 5b). Most importantly, the sensitivity of cutaneous C-fiber and Aδ-fiber nociceptors from TMEM100 KO mice was not altered by intra-articular CFA injection (Fig. 6c and Supplementary Fig. 5b).

In addition to von Frey thresholds we also examined the effect of CFA-treatment on the supra-threshold firing patterns of C- and A-fiber nociceptors. To this end, we applied a series of ramp-and-hold stimuli of increasing magnitude to the receptive fields using a piezoelectric micromanipulator. These experiments showed that cutaneous C-fibers from CFA-treated mice fired significantly more action potentials in response to suprathreshold mechanical stimuli (Fig. 6d, e). Aδ-fiber nociceptors also showed a trend towards higher firing rates, which was, however, only significant for the strongest tested stimulus (Supplementary Fig. 5a, c). Strikingly, the mechanical activation thresholds and the firing rates of both C-fiber nociceptors and Aδ-fiber

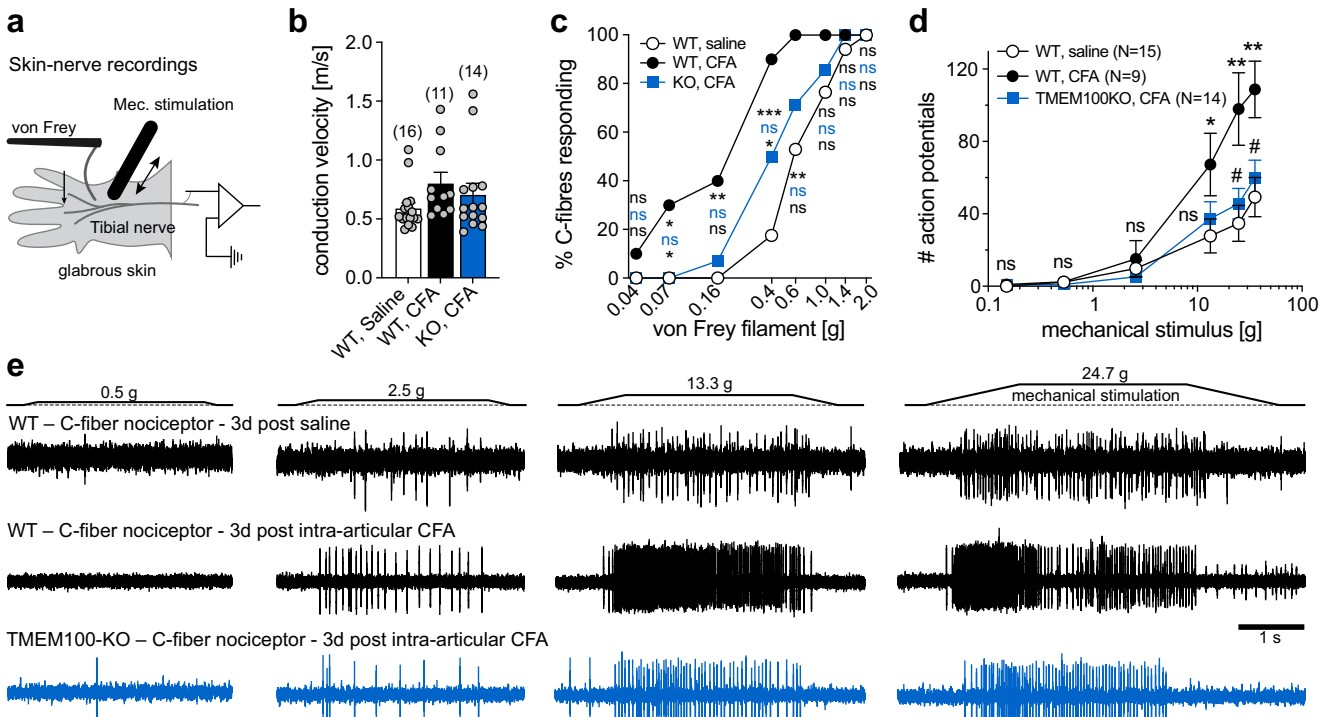

**Fig. 6 | Sensitization of cutaneous C-fiber nociceptors contributes to secondary mechanical hypersensitivity. a** Cartoon depicting the experimental approach. **b** Scatter dot plots of the conduction velocities [m/s] of the examined afferent fibers from the indicated mice. Bars represent means ± SEM. **c** Comparison of the proportions of C-fiber nociceptors that respond to mechanical stimulation with the indicated von Frey filaments. The proportions were calculated from all C-fibers (same N-numbers as in b) recorded from 3 different mice per group and were compared pairwise using the two-sided Chi-square test. *P*-values (ns, *P* > 0.05; **P* < 0.05; ***P* < 0.01; ****P* < 0.001) are provided next to the symbols in the graph and refer to WT-saline vs. WT-CFA (top, black), WT-saline vs. TMEM100KO-CFA (middle, blue) and WT-CFA vs. TMEM100KO (bottom, black). Exact *P*-values are provided

together with additional statistical information in the source data file.
**d** Comparison of the firing rates evoked by a series of ramp-and-hold stimuli with increasing amplitudes that exerted the indicated force to the receptive fields. Symbols represent the mean ± SEM numbers of action potentials, which were compared using multiple two-sided Mann-Whitney tests. (WT-saline vs. WT-CFA: **P* < 0.05; ***P* < 0.01; WT-CFA vs. TMEM100KO-CFA: #*P* < 0.05). Exact *P*-values are provided together with additional statistical information in the source data file.
**e** Example traces of mechanically-evoked action potentials recorded from single nerve fibers from the tibial nerve of WT mice 3 days post intraarticular saline (top), WT mice 3 dpi CFA (middle) and TMEM100KO mice 3 dpi CFA (bottom). Source data are provided as a Source Data file.

nociceptors from CFA-treated TMEM100KO mice were indistinguishable from those of saline-treated WT control mice (Fig. 6d, e and Supplementary Fig. 5a, c).

Taken together, these results show that CFA-induced knee joint inflammation shifts the mechanosensitivity of a large subset of cutaneous C-fiber nociceptors towards innocuous mechanical stimuli, which correlates with the leftward shift of the paw withdrawal thresholds in the same mice and therefore suggests that the sensitization of cutaneous C-fiber and Aδ-fiber nociceptors also contributes to secondary mechanical hypersensitivity in the hind paw induced by knee joint inflammation.

**Overexpression of TMEM100 in articular afferents induces secondary allodynia in the hind paw but no knee joint pain**
The observation that inhibition of un-silencing of MIAs by knocking out TMEM100 prevents the development of long-lasting secondary mechanical hypersensitivity, suggested that the primary function of MIAs might be to induce secondary mechanical hypersensitivity. To test this hypothesis, we next un-silenced articular MIAs without inducing an inflammation. To this end, we selectively overexpressed TMEM100 in knee joint afferents by intra-articular injection of an AAV-PHP.S-TMEM100-Ires-dsRed virus (30 μl, 1.5 × 10[11] vg; Fig. 7a). Four days after intraarticular AAV-PHP.S-TMEM100-Ires-dsRed administration, we observed prominent dsRed fluorescence in a total of 339 ± 7 neurons in ipsilateral L3 and L4 DRG (Fig. 7a), which−considering that TMEM100 and dsRed are coupled with an IRES cassette−indicates TMEM100 overexpression. Most importantly, we observed numerous

dsRed expressing nerve fibers in the saphenous nerve proximal to the knee (Fig. 7b), which includes the medial articular nerve that supplies the knee joint and in which silent nociceptors had first been described[11], but hardly any dsRed+ fibers in the tibial nerve distal to the knee, which contains cutaneous afferents that supply the plantar surface of the hind paw (Fig. 7b). Hence, intraarticularly administered AAV-PHP.S-TMEM100-Ires-dsRed causes selective overexpression of TMEM100 in knee joint afferents, but not in afferents that supply the plantar surface of the hind paw. Interestingly, TMEM100-overexpressing mice exhibited normal gait, indicating that un-silencing of knee joint MIAs does not trigger knee joint pain (Fig. 7c). Strikingly, however, these mice developed profound mechanical hypersensitivity in the ipsilateral hind paw five days post AAV injection, which persisted until the end of the observation period (21 dpi; Fig. 7d). Thus, the mechanical paw withdrawal thresholds decreased from 0.92 ± 0.066 g (−1 dpi) to 0.373 ± 0.035 g (14 dpi). Mice that received a control virus without TMEM100 (AAV-PHP.S-dsRed) did not show any signs of mechanical hypersensitivity. We also monitored other behaviors that might be indicative of pain using the LABORAS system, but did not observe any significant differences between TMEM100-overexpressing and control mice (Supplementary Fig. 6).

In accordance with the behavioral outcome of TMEM100 overexpression in articular afferents, single-unit action potential recordings from a tibial nerve-glabrous skin preparation showed that cutaneous C-fiber nociceptors fire about twice as many action potentials in response to supra-threshold stimuli and have significantly reduced von Frey thresholds in mice that overexpress TMEM100 in articular

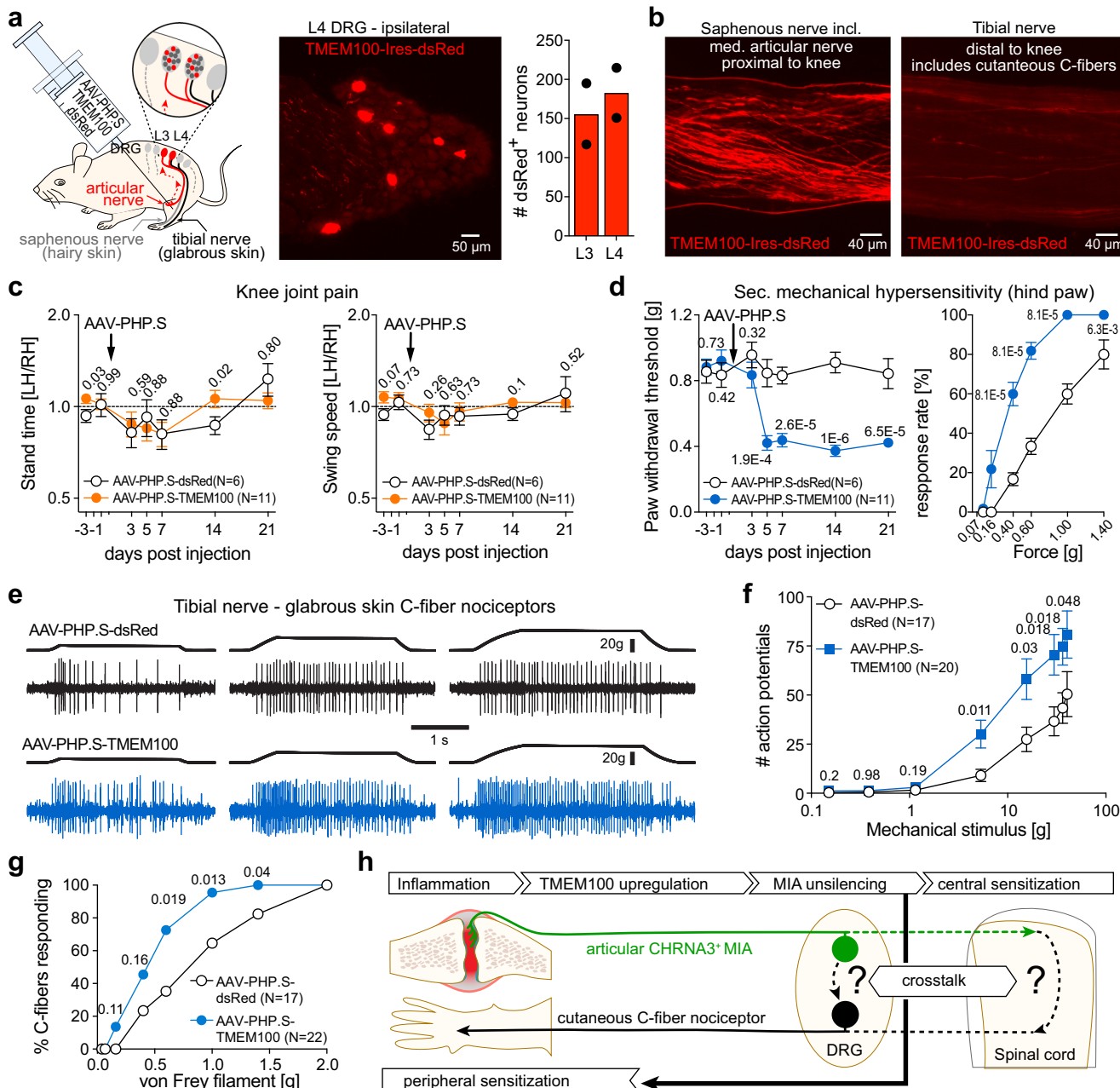

**Fig. 7 | Overexpression of TMEM100 in articular afferents induces secondary mechanical hypersensitivity in the hind paw but no knee joint pain. a** Cartoon depicting the experimental approach (left), image of a DRG section from a mouse that had received intraarticular AAV-PHP.S-TMEM100-Ires-dsRed (middle) and quantification of the mean ± SEM number of dsRed⁺ neurons (right) in L3 and L4 DRGs from two mice. **b** dsRed fluorescence in the tibial nerve distal to the knee (right) and the saphenous (left) nerve proximal to the knee which contains the medial articular nerve. **c** Comparison of the mean ± SEM stand time (left) and leg swing speed ratios (right) of WT mice at the indicated timepoints after i.a. injection of AAV-PHP.S-dsRed control virus (white circles) and AAV-PHP-S-TMEM100-Ires-dsRed (orange circles). Ratios were compared using multiple two-sided Mann-Whitney tests (P-values and N-numbers are provided next to the symbols and in the graph legend). **d** Comparison of the mean ± SEM mechanical paw withdrawal thresholds (left) and the response rates at 14 dpi (right, % paw withdrawals in response to five successive stimulations with the indicated von Frey filaments) of

the same mice as in **c**, using two-sided Mann-Whitney test (P-values are provided next to the symbols; number of tested animals is the same in both panels and is indicated in the graph legend). **e** Example traces of mechanically-evoked action potentials recorded from cutaneous C-fiber nociceptors in the tibial nerve from control mice (top, AAV-PHP.S-dsRed) and from mice that overexpress TMEM100 in articular afferents (bottom, AAV-PHP.S-TMEM100-Ires-dsRed). **f** Comparison of the firing rates evoked by ramp-and-hold stimuli that exerted the indicated force to the receptive fields. Symbols represent means ± SEM numbers of action potentials, which were compared using multiple two-sided Mann-Whitney tests (exact P-values are provided next to the symbols). **g** comparison of the proportions of C-fiber nociceptors that respond to mechanical stimulation with the indicated von Frey filaments. The proportions were compared pairwise using two-sided Chi-square test (exact P-values are provided next to the symbols). **h** cartoon depicting the proposed mechanism underlying the induction of secondary hypersensitivity. Source data are provided as a Source Data file.

afferents compared to mice that had received a control virus (Fig. 7e–g). Similar to CFA-treatment, TMEM100 overexpression in articular afferents also reduced the mechanical activation thresholds of cutaneous Aδ-fiber nociceptors and increased the action potential

firing rate in response to suprathreshold stimuli (Supplementary Fig. 7).

In summary, our data shows that TMEM100 overexpression-induced un-silencing of mechanically insensitive articular afferents is

sufficient to trigger mechanical hypersensitivity in remote skin regions (Fig. 7d). Together with the observation that TMEM100 is specifically up-regulated in MIAs during CFA-induced inflammation and that knock-out of TMEM100 exclusively abolishes long-lasting secondary mechanical hypersensitivity, these results suggest that sensory input from unsilenced MIAs is the main trigger for the induction of secondary mechanical hypersensitivity (Fig. 7h).

## Discussion

The peculiar properties of MIAs have fueled speculations about a prominent contribution to inflammatory pain ever since they had first been described more than thirty years ago[11,17,42], but hitherto, neither the molecular mechanism underlying their un-silencing, nor their exact role in pain signaling have been deciphered.

With regards to the molecular mechanism of MIA un-silencing, we have previously shown that in-vitro NGF-induced un-silencing requires de-novo gene transcription and that mechanosensitivity in MIAs is mediated by the mechanically-gated ion channel PIEZO2, the expression of which is, however, not changed during un-silencing (Fig. 1d)[19]. Here we show that TMEM100 is specifically up-regulated in CHRNA3-EGFP+ MIAs during inflammation (Figs. 1d, e and 3c) and demonstrate that over-expression of TMEM100 is sufficient to un-silence MIAs in-vitro (Fig. 1f–h), whereas knock-out of TMEM100 prevents un-silencing in a mouse model of CFA-induced monoarthritis (Fig. 5b, d). Considering that CHRNA3-EGFP+ MIAs express high levels of PIEZO2 even in conditions in which they are not mechanosensitive[19] (Fig. 1d) and that PIEZO2 does not require the presence of an auxiliary subunit for normal function, we hypothesize that PIEZO2 is somehow kept 'silent' in these neurons under normal conditions and is primed by the up-regulation of TMEM100. CHRNA3-EGFP+ MIAs express four known PIEZO2 inhibitors, namely MTMR2[23], SERCA2 (Atp2a2)[27], Annexin A6 (Anxa6)[26] and Nedd4-2[28], but since neither of them is down-regulated by NGF (Fig. 1sd) and all four are ubiquitously expressed in sensory neurons[2], it seems highly unlikely that any of them is involved in keeping PIEZO2 silent in MIAs. Hence, while our study identified TMEM100 as a key protein that is sufficient and required for the un-silencing of MIAs, the mechanism by which PIEZO2 is kept inactive in CHRNA3-EGFP+ MIAs remains elusive.

TMEM100 had originally attracted our attention because it was previously shown to enhance TRPA1 activity in a TRPV1-dependent manner[29] and because there is ample evidence supporting a role of TRPA1 in fine-tuning mechanosensitivity of sensory afferents. Thus, it was shown that pharmacological blockade as well as knock-out of TRPA1 partially inhibit mechanotransduction currents in cultured DRG neurons and reduce the firing rate of cutaneous C-fiber nociceptors[43–46]. Since TRPA1, to the best of our knowledge, is not activated by mechanical indentation of the cell membrane but is directly gated by intracellular calcium[47], a possible mechanistic explanation for the role of TRPA1 is that it increases mechanosensitivity of sensory neurons by amplifying PIEZO2-mediated $Ca^{2+}$ influx. Consistent with the findings of Weng and colleagues[29], we found that CFA-induced knee joint inflammation enhances TRPA1 activity in a TMEM100-dependent manner (Figs. 3h and 5e). Most importantly, our data substantiate and extend the findings of Weng et al. (2015), by showing that TMEM100 is exclusively up-regulated in CHRNA3-EGFP+ neurons—at least in CFA-induced knee monoarthritis and after in-vitro NGF treatment (Figs. 1d, e and 3c)—and accordingly only enhances TRPA1 activity in MIAs but not in other nociceptors (Figs. 3h and 5e). Hence our data suggests that TMEM100 is a pleiotropic protein that on the one hand induces mechanosensitivity in MIAs by priming PIEZO2 via a yet unknown mechanism and on the other hand amplifies PIEZO2-dependent mechanosensitivity by releasing TRPA1 from inhibition by TRPV1.

With regards to the physiological role of MIAs in pain signaling, our data suggests that their primary function is to trigger secondary mechanical hypersensitivity. Pain hypersensitivity usually manifests as primary hypersensitivity (hyperalgesia or allodynia), which is restricted to the site of inflammation and is thought to result from peripheral sensitization—i.e., direct sensitization of nerve endings in the inflamed or injured tissue—and usually resolves concurrently with the initial cause of pain. In addition, patients often develop secondary mechanical and thermal pain hypersensitivity that spreads beyond the site of inflammation or injury. This secondary pain hypersensitivity often persists after the initial cause for pain has resolved and clinical data indicate that the area and intensity of secondary pain hypersensitivity correlates with the likelihood of developing chronic pain[48]. Secondary pain hypersensitivity has been observed in numerous pain disorders and rodent models of pain, including experimentally induced arthritis, and is thought to result from central sensitization—i.e., a strengthening of synaptic transmission in pain processing circuits in the spinal cord[40,41,49–52]. The peripheral inputs that trigger central sensitization and hence secondary mechanical hypersensitivity are, however, still unknown.

Here we show that blocking the un-silencing of articular MIAs by knocking out TMEM100 prevents the development of long-lasting secondary mechanical hypersensitivity in remote skin regions, but does not alter pain at the actual site of CFA-induced inflammation (Fig. 4). Moreover, TMEM100 overexpression-induced un-silencing of knee joint MIAs in the absence of inflammation or injury, induces mechanical hypersensitivity in the paw but not pain hypersensitivity in the knee joint (Fig. 7c, d), which indicates that secondary mechanical hypersensitivity does not result from spreading of inflammation or from sensitization of cutaneous nerve fibers passing through the inflamed joint region. Finally, our skin-nerve recordings demonstrate that the sensitization of cutaneous C-fiber nociceptors (Figs. 6 and 7e–g), in addition to the previously described central sensitization, also contributes to the development of secondary mechanical hypersensitivity. Considering that the un-silencing of knee joint MIAs by selective overexpression of TMEM100 is sufficient to induce sensitization of cutaneous afferents that innervate remote skin regions, we hypothesize that sensitization is induced by a yet unknown—probably central—mechanism (e.g., crosstalk at the level of the spinal cord; see Fig. 7h).

The interpretation of our behavioral data relies on the assumption that the Catwalk XT gait analysis system measures primary knee joint pain. Some people might argue that pain in the paw might also alter gait and, on the other hand, that changes in gait may not necessarily reflect joint pain but could also be altered by structural damage of the joint. The observations that (i) TMEM100 overexpression dramatically reduces the paw withdrawal thresholds but does not alter gait in the catwalk XT assay and that (ii) the time courses of changes in gait parameters and in paw withdrawal thresholds differ, however, argue against the possibility that secondary mechanical hypersensitivity in the paw confounds the outcome of the catwalk XT assay. Structural damage of the joint also seems unlikely, because changes in gait and stride parameters are only transient and return to baseline values within a few days.

The interpretation of the AAV-mediated TMEM100 overexpression experiments is also complex, because intra-articularly administered AAV-PHP.S infects all subclasses of sensory afferents in the knee joint. A sensitization of non-MIAs by TMEM100 overexpression, however, seems unlikely because if that had been the case then we should have observed knee joint pain in addition to secondary mechanical hypersensitivity, which was not the case (Fig. 7c). Moreover, the observation that TMEM100 expression is exclusively upregulated in MIAs in CFA-induced monoarthritis (Fig. 3c) and that mice lacking TMEM100 only show functional deficits in MIAs but not in other articular nociceptors (Figs. 3 and 5) and selectively lose secondary mechanical hypersensitivity but not knee joint pain, strongly support a specific role of MIAs in inducing secondary mechanical hypersensitivity.

Thus, taken together, our data support a mechanistic model of inflammatory knee joint pain in which polymodal nociceptors signal primary hyperalgesia, while MIAs are responsible for triggering secondary mechanical hypersensitivity. Our data shows that inflammation-induced upregulation of TMEM100 un-silences MIAs, which subsequently—via a yet unknown central mechanism—triggers sensitization of cutaneous nociceptor thereby leading to secondary mechanical hypersensitivity in skin regions remote from the site of inflammation (Fig. 7h). We have not explicitly tested if un-silencing of MIAs also triggers central sensitization, which is known to contribute to secondary mechanical hypersensitivity. Yet, considering that the AAV-PHP.S-TMEM100 induced reduction of paw withdrawal thresholds (Fig. 7d) was larger than the reduction of the mechanical activation thresholds of individual cutaneous C- and Aδ-fiber nociceptors (Fig. 7g and Supplementary Fig 5b), it seems highly likely that central processing of nociceptor input was also altered by the un-silencing of articular MIAs such that subliminal nociceptor-derived inputs evoked pain.

By demonstrating that primary and secondary pain hypersensitivity are triggered by separate subclasses of primary sensory afferents and considering that MIAs constitute almost fifty percent of all nociceptors in viscera and deep somatic tissues, our study provides an invaluable framework for future studies that aim at deciphering the contribution of different afferent subtypes to other clinically relevant forms of pain and to develop new strategies for preventing the transition from acute to chronic of pain after injury, inflammation or surgical interventions.

## Methods

### Animals

All experiments were conducted in accordance with the European Communities Council Directive (EU and institutional guidelines) including the ethical guidelines of 'Protection of Animals Act' under supervision of the 'Animal Welfare Officers' of Heidelberg University and were approved by the local governing body (Regierungspraesidium Karlsruhe, approval number G16/20). ARRIVE guidelines were followed and sample sizes were calculated as previous experience with G-power analyses.

CHRNA3-EGFP mice (Tg(Chrna3-EGFP)BZ135Gsat/Mmnc) were purchased from the Mutant Mouse Resource & Research Center (MMRRC) and were backcrossed to a C57Bl/6 J background. Conditional nociceptor TMEM100 knock-out mice were generated by crossing mice that carry a conditional allele for TMEM100 (B6.Tmem100tm1.1Yjl)[37] with SNS-Cre mice C57BL/6-Tg(SCN10A-Cre)1Rkun/Uhg[38] (gift from Rohini Kuner). To enable identification of MIAs, these mice were further crossed with CHRNA3-EGFP mice. Furthermore, to identify different nociceptor subclasses for RT-qPCR experiments, Tg(Npy2r-cre)SM19Gsat/Mmucd x B6;129S-Gt(ROSA)26Sortm32(CAG-COP4*H134R/EYFP)Hze/J (Npy2rCre;ChR2-EYFP) mice, in which Aδ-fiber nociceptors express EYFP were used[6]. Mice were maintained in the Interfaculty Biomedical Facility of Heidelberg University according to institutional guidelines on a 12/12-h light-dark cycle in an enriched housing environment and had access to food and water ad libitum at 21 °celsius and 50% humidity. For all experiments only adult (age 8–15 weeks) male and female mice were used. Behavioral experiments were conducted at the Interdisciplinary Neurobehavioral Core of Heidelberg University. Prior to the start of experiments animals with the same genetic background and age were randomly assigned to the different experimental groups. To reduce bias investigators were blinded to group identity including treatment (CFA/Saline) and genotype (TMEM100KO/WT).

### HEK293 cell maintenance and transfection

To assess the mechanosensitivity of TMEM100 and of PIEZO2 in the presence and absence of TMEM100, the constructs were (co)-transfected into HEK293 cells using the calcium phosphate method. Cells were grown in DMEM (Thermo Fisher) supplemented with 10% FBS (Thermo Fisher), 2 mM L-glutamine (Thermo Fisher) and penicillin streptomycin (Thermo Fisher, 100 U/mL) at 37 °C and 5% $CO_2$. The day before transfection, cells were seeded on poly-L-lysine treated glass coverslips. For transfection, growth medium was replaced with transfection medium consisting of DMEM, 10% calf serum (Thermo Fisher) and 4 mM L-Glutamine. DNA (0.6 μg/coverslip) was diluted in 100 μl water and after adding CaCl2 2.5 M (10 μL per coverslips) the solution was vigorously mixed. Then, 2 x BBS (in mM, 50 HEPES, 280 NaCl, 1.5 Na2HPO4, pH 7.0; 100 μL/coverslip) was added and vortexed. The resulting DNA mix was added to the transfection medium. After 3–4 h at 37 °C (5% CO2) the transfection medium/DNA mix was replaced with regular HEK293 growth medium. PIEZO2 function was assessed 48 h after transfection.

### Primary DRG cell culture

For primary DRG cultures, mice were sacrificed by placing them in a $CO_2$-filled chamber for 2–4 min followed by cervical dislocation. Lumbar L3 and L4 DRG were collected in Ca2+ and Mg2+-free PBS. DRG were subsequently treated with collagenase IV for 30 min (0.5 mg/ml, Sigma-Aldrich, C5138) and with trypsin (0.5 mg/ml, Sigma-Aldrich, T1005) for further 30 min, at 37 °C. Digested DRG were washed twice with growth medium [DMEM-F12 (Gibco®, Thermo Fisher Scientific) supplemented with L-glutamine (2 μM, Sigma-Aldrich), glucose (8 mg/ml, Sigma-Aldrich), penicillin (200 U/ml)–streptomycin (200 μg/ml) (Gibco®, Thermo Fisher Scientific), 5% fetal horse serum (Gibco®, Thermo Fischer Scientific)], triturated using a pipette with filter tips of decreasing diameter (10× up and down with 1000 μl filter tip, 10× up and down with 200 μl filter tip) and plated in a droplet of growth medium on a glass coverslip precoated with Laminin (GG-12-Laminin coated coverslips, Neuvitro). To allow the dissociated neurons to adhere, coverslips were incubated for 3 h at 37 °C in a humidified 5% incubator before being flooded with fresh growth medium. Depending on the experiment neurons were used directly after adding fresh growth medium (see Reverse transcription and quantitative real-time PCR) or after 24 h of incubation (see Patch-clamp recordings). For the experiments shown in Fig. 1f–h, primary DRG cultures were transfected using the Amaxa Nucleofector 4D (Lonza) following the manufactures instructions.

### Inflammatory knee pain model

To induce inflammatory knee joint pain the Complete Freund's Adjuvant (CFA)-induced knee joint monoarthritis model was used[53]. In brief, animals were anesthetized in a transparent plexiglass chamber filled with 4% isoflurane in 100% O2 at a flow rate of 1.0 L/min for 3 min. During the procedure anesthesia was maintained using a nosecone delivering a 1.5% Isofluran-O2 mixture while respiratory function was monitored carefully. Adequate anesthesia was confirmed by absence of the pedal reflex (toe pinch). Then, ophthalmic ointment was applied to both eyes to prevent desiccation and the animals were placed in a supine position. Prior to injection of CFA, the left knee was shaved using a commercially available electrical facial hair trimmer, disinfected with povidone-iodine scrub (7.5% solution, Braunol®) and stabilized in a bent position by placing the index finger beneath the knee joint and the thumb above the anterior surface of the ankle joint. The patellar tendon shining through the shaved skin served as visual landmark for the injection. To ensure a precise intraarticular (i.a.) injection the gap inferior to the lower edge of the patella was identified by running a 30 G Insulin syringe horizontally along the knee. To mark the injection level gentle pressure was applied to skin without piercing it leaving behind a horizontal dermal print line. For the injection the needle was lifted vertically at the marked level and inserted at the midline through the patella tendon perpendicular to the tibial axis. Then, the needle was advanced approximately 2–2.5 mm without resistance to fully enter the knee joint and 30 μl of CFA (1 μg/μl) or saline were injected into the joint cavity. After the procedure, the

injection site was disinfected and the knee was briefly massaged and mobilized to ensure even distribution of CFA/saline before the animals were returned to their home cages placed on a heating pad for recovery.

## Behavioral testing

All behavioral tests were performed in awake, unrestrained mice by experienced investigators blinded to group identity. Before testing, all animals were habituated to the behavioral test setups at least 3 times over a total of 3 days (1×/d per setup) in the week before starting the behavioral experiment. For the von Frey (vF) and Hargreaves' test animals were habituated for 1 h/setup, and 30 to 60 min immediately before each test with the experimenter present in the same room. For the CatWalk XT a habituation session was completed when mice voluntarily crossed the runway 3 times without stopping, turning around, or changing direction (approx. 5 min/animal). On testing days, acclimatization to the CatWalk setup was not necessary. Behavioral assays were always carried out in the same order (CatWalk, vF, Hargreaves) using the same rooms and same test setups at the same time point of the day between 8 a.m and 3 p.m. Prior to knee injection, at least two baseline measurements on two different days were conducted for all behavioral tests. After injections, behavior was evaluated 1, 2, 3, 5 and 7 days post injection (dpi) and then in weekly intervals (14 dpi) for a total of 3 weeks (21dpi).

## Von Frey Test: mechanical sensitivity

Mechanical sensitivity was assessed using the von Frey test[54]. Animals were placed in transparent plastic chambers (Modular animal-enclosure; Ugo Basile Srl, Gemonio, Italy) on a 90 × 38 cm perforated metal shelf (Framed testing surface; Ugo Basile Srl, Gemonio, Italy) that was mounted on a stimulation base. The plantar surface of the animals' hind paws was perpendicularly stimulated with graded von Frey filaments (Aesthesio® Precision Tactile Sensory Evaluators) of different forces, ranging from 0.07 to 1.4 g without moving the filaments horizontally during application. Each filament was applied 5 times to both the right and left hind paws and the response rate to stimulation in percent (positive response/number of applied stimuli) was used to express mechanical sensitivity. Withdrawal of the stimulated paw was defined as a positive response. Between stimulations of the same hind paw animals had least 1 min break. Prior to knee injection baseline withdrawal frequencies were determined by measuring the withdrawal response rates for all filaments on two different days. After injections, mechanical sensitivity was evaluated 1, 2, 3, 5, and 7 days post injection (dpi) and then in weekly intervals for a total of 3 weeks (21dpi). The 50% withdrawal threshold (WDT) in grams was determined by fitting the response rate vs. von Frey force curves with a Boltzmann sigmoid equation with constant bottom and top constraints equal to 0 and 100, respectively.

## Hargreaves' test: thermal sensitivity

Thermal sensitivity was assessed according to Hargreaves' method[55] using the Plantar test (Hargreaves Apparatus; Ugo Basile Srl). In brief, withdrawal latency (WDL) in seconds to an infrared (IR) heat beam stimulus applied to the plantar surface of the hind paws was recorded to determine thermal sensitivity. The IR intensity of the radiant heat source was adjusted to obtain baseline WDL between 5 and 7 s (IR intensity 50%), with a pre-determined cut-off time of 15 s to prevent tissue damage. Each paw was assessed three times and between trials for the same paw, animals had at least an one minute break. To prevent an order effect the paw testing order was chosen randomly. Baseline and post-interventional measurements were conducted at the same time points as described for the vF test.

## CatWalk XT: gait analysis

To quantitatively assess locomotion, the CatWalk XT (version 10.6) gait analysis system (Noldus, Netherlands system) was used. This system consists of an enclosed black corridor (1.3 m length) on a glass plate. Inside this glass plate a green LED light is internally reflected. When animals touch the glass plate the light is refracted on the opposite side so that areas of contact become illuminated and detectable. Using the Illuminated Footprints™ technology, videos including the illuminated areas (e.g., paw prints) can be recorded using a high speed color camera (100 frames/s) that is positioned underneath the glass plate. The data is automatically transferred to computer running the CatWalk XT software for further gait analysis. Animals were habituated to the set up as described above (see Behavioral testing). On testing days, animals were placed on one end of the corridor and were allowed to transverse it voluntarily without any external enforcement after setting up the walkway according to the manufacturer's recommendations. For each mouse on each measurement time point three compliant runs were recorded. A compliant run was defined as a mouse walking across the runway without stopping, turning around, or changing direction and meeting the following pre-determined run criteria: minimum run duration of 0.5 s and a maximum run duration of 12 s. For all runs the same detection settings were used (camera gain: 16.99, green intensity threshold: 0.10, red ceiling light: 17.7, green walkway light: 16.5). In our analyses we focused on the following gait parameters: stand time in seconds, paw print area in cm², swing speed in cm/s. To better illustrate pain-associated changes in the gait cycle including the stand and swing phase, run data of the left (LH) and right hind paw (RH) are displayed as ratio (LH/RH). For each testing day the ratios (LH/RH) of all three runs per animal were averaged so that the mean of three compliant runs represents the overall result of the animal on that testing day.

## Homecage monitoring

The LABORAS (Laboratory Animal Behavior Observation, Registration, and Analysis System) is a monitoring tool used to observe and analyze animal behavior. It utilizes a carbon fiber platform to detect unique vibrations produced by the animal as it moves around in its homecage. The platform is equipped with a software (LABORAS software version 2.6) that processes these vibrations into various behavioral parameters, such as locomotion (e.g.,), immobility, rearing, drinking and grooming. These parameters were calculated as frequency counts. Animals were placed individually in a calibrated cage under standard housing conditions with free free access to food and water. Their activity was continuously monitored for 16–24 h before and 3 or 18 days after an intraarticular knee joint injection with Saline/CFA or AAV-PHP.S-TMEM100-Ires-dsRed/AAV-PHP.S-dsRed, respectively.

## Retrograde labeling

To identify sensory neurons innervating the knee joint retrograde labeling with Fast Blue (FB, #17740-1, Polysciences) was performed. To this end, the same anesthesiological and surgical approach was used as described above (see Inflammatory knee pain model). A total of 2 μl of a 4% FB (in saline) solution were injected i.a. in both knee joints using a 10 μl Hamilton syringe fitted with 30 G needle. After a waiting period of 7d allowing the FB to retrogradely travel to the DRG, animals were further processed depending on the following experiment. To quantify the knee innervating neurons (Fig. 3a), animals were sacrificed for microscopy (see Tissue processing and immunochemistry). For electrophysiological and qPCR experiments, the left knee was injected again, but with CFA as described above (see Inflammatory knee pain model) and 3 days thereafter—at the time of maximum pain – the animals were euthanized for primary DRG cultures.

## Patch-clamp recordings

Whole cell patch clamp recordings were made from retrogradely FB-labeled sensory neurons innervating the knee (see Retrograde labeling). To distinguish between MIAs and other peptidergic nociceptors, CHRNA3-EGFP⁺/FB⁺ neurons and small (<30 μm) IB4⁻/FB⁺ neurons (see

Immunohistochemistry) were recorded, respectively. These two sub-populations account for the majority of nociceptive knee joint afferents (see Fig. 3b). Cells from both WT and TMEM100KO animals after CFA and control treatment were assessed. To this end, 7d after retrograde labeling of the DRG from both knees animals received a second injection into the left knee using CFA or Saline. 3d after this second injection, at the time of maximum CFA-induced pain behavior, the animals were sacrificed. L3 and L4 DRG from the ipsi- (CFA/Saline) and contralateral side were collected separately and cultured for 16–24 h (see Primary DRG culture) until used for whole cell patch clamp recordings.

Whole cell patch clamp recordings were made at room temperature (20–24 °C) using patch pipettes with a tip resistance of 2–4 MΩ that were pulled (Flaming-Brown puller, Sutter Instruments, Novato, CA, USA) from borosilicate glass capillaries (BF150-86-10, Sutter Instrument). The patch pipettes were filled with a solution consisting of 110 mM KCl, 10 mM NaCl, 1 mM MgCl$_2$, 1 mM EGTA, 10 mM HEPES, 2 mM guanosine 5′-triphosphate (GTP) and 2 mM adenosine 5′-triphosphate (ATP) adjusted to pH 7.3 with KOH. The bathing solution contained 140 mM NaCl, 4 mM KCl, 2 mM CaCl$_2$, 1 mM MgCl$_2$, 4 mM glucose, 10 mM HEPES and was adjusted to pH 7.4 with NaOH. All recordings were made using an EPC-10 amplifier (HEKA, Lambrecht, Germany) in combination with Patchmaster© and Fitmaster© software (HEKA). Pipette and membrane capacitance were compensated using the auto function of Patchmaster and series resistance was compensated by 70% to minimize voltage errors. Mechanically activated currents were recorded in the whole-cell patch-clamp configuration. Neurons were clamped to a holding potential of −60 mV and stimulated with a series of mechanical stimuli in 0.8 μm increments with a fire-polished glass pipette (tip diameter 2–3 μm) that was positioned at an angle of 45° to the surface of the dish and moved with a velocity of 3 μm/ms by a piezo based micromanipulator called nanomotor© (MM3A, Kleindiek Nanotechnik, Reutlingen, Germany). The evoked whole cell currents were recorded with a sampling frequency of 200 kHz. Mechanotransduction current inactivation was fitted with a single exponential function $(C1 + C2*exp(-(t-t0)/\tau_{inact})$, where C1 and C2 are constants, t is time and $\tau_{inact}$ is the inactivation time constant[56].

## Tissue processing and immunochemistry

To quantify retrogradely labeled neurons, DRG were dissected in ice-cooled PBS, fixed with Zamboni´s fixative for 1 h at 4 °C and incubated overnight in 30% sucrose at 4 °C. Then, DRG were embedded in optimum cutting temperature compound (Tissue-Tek™ O.C.T. Compound; Sakura Finetek Germany GmbH, Staufen), cut into 16 μm cryo-sections using a cryostat (Leica CM1950, Leica, Wetzlar, Germany) and mounted onto slides (Microscope Slides SUPERFROST PLUS; Thermo Fisher Scientific, Schwerte, Germany) which were stored at −80 °C until used for immunohistochemistry. After drying, sections were treated with 50 mM Glycine in PBS for 20 min, washed twice with 0.2% Triton X-100 in PBS (0.2% PBST), blocked with 10% normal donkey serum (NDS) and 1% bovine serum albumin (BSA) in 0.2% PBST for 30 min and then incubated with primary antibodies overnight at 4 °C. Primary antibodies were diluted in the blocking solution (10% NDS and 1% BSA in 0.2% PBST). Next day, sections were washed 4 × 15 min with 0.2% PBST, subsequently incubated with secondary antibodies for 1 h at room temperature (RT), washed with 0.2% PBST four times (15 min each), dried and coverslipped with FluoroGel (FluoProbes®, Interchim, Montluçon France).

Cultured DRG neurons (see Primary DRG cultures) for electrophysiological and qPCR experiments were counterstained with Alexa Fluor™ −568 conjugated IB4 (2.5 μg/ml, Isolectin GS-IB from Griffonia simplicifolia, Alexa Fluor™ 568 Conjugate, Invitrogen™/Thermo Fischer Scientific, I21412) for 10–15 min at room temperature to identify different nociceptor subpopulations (CHRNA3-EGFP$^+$/FB$^+$ and IB4$^-$/FB$^+$ neurons).

Knees were dissected in cold PBS and fixed with Zambonis fixative overnight. Then, the knee joints were washed in purified water (Milli-Q, Merck KGaA, Darmstadt, Germany) for 3 × 30 min before being decalcified by submerging the samples in 10% EDTA in PBS for 7–10 days (PBS/EDTA was replaced every other day) on a tube roller mixer at 4 °C. After decalcification the samples were washed in PBS for 3 × 10 min at RT and cryoprotected in 30% sucrose solution at 4 °C for at least 24 h. For the preparation of tissue sections, samples were embedded in optimum cutting temperature compound (Tissue-Tek™ O.C.T. Compound; Sakura Finetek Germany GmbH, Staufen), cut into 25 μm consecutive coronal cryo-sections in arterior-posterior direction and mounted onto microscope slides (SUPERFROST PLUS; Thermo Fisher Scientific, Schwerte, Germany). After drying at RT for 1 h, sections were incubated with 50 mM Glycine in PBS for 30 min, washed and blocked 3 × 10 min with 0.5% Tween® 20 in PBS (0.5% PBS-Tw) and then incubated with primary antibodies for 3d at 4 °C. Primary antibodies were diluted in a PBS solution containing 1% BSA and 0.3% Triton X-100. Sections were then washed 3 × 10 min with 0.5% PBS-Tw and subsequently incubated for 2 h at RT with secondary antibodies diluted in PBS with 1% BSA and 0.3% Triton X-100. Finally, the slides were washed with PBS (3 ×10 min), dried and coverslipped using FluoroGel mounting medium with 4,6-diamidino-2-phenylindole (DAPI) counter stain (FluoProbes®, Interchim, Montluçon FRANCE).

Immunostaining images were captured with the Nikon DS-Qi2 camera mounted on a Nikon Ni-E epifluorescence microscope using appropriate filter cubes and identical exposure times for all slides within one experiment.

Silent afferent density (EGFP$^+$/CGRP$^+$ fibers) was quantified in anatomical regions including FP (Hoffa's fat pad), LM (lateral meniscus), MM (medial meniscus), LJC (lateral joint capsule), MJC (medial joint capsule) and CL (cruciate ligament) defined according to previously described landmarks[57] using the area fraction tool of NIH ImageJ software (ImageJ 1.53e; Java 1.8.0_172 [64-bit]). In brief, after conversion to 8-bit and background subtraction, image auto local thresholds were set using the Bernsen method. Then, the different immunostaining images (channels) were merged and processed using the image calculator tool to display only double positive (EGFP$^+$/CGRP$^+$) signals. Finally, the predefined anatomical regions of interest were overlaid and the area fraction determined representing the labeling density of silent afferents per anatomical region in percent. For each animal, at least three photomicrographs per anatomical region were analyzed and averaged. The labeling density per anatomical region for all animals was expressed as mean ± SEM. For illustration purposes (Fig. 2a) representative images of coronal 100 μm knee sections (Cryostat) were acquired and stitched using a Leica SP8 Confocal microscopy platform equipped with a Lasx 3.5. Laser, detector powers were optimized for the combination of antibodies.

## Antibodies

The following primary antibodies were used: rat anti-GFP (Nacalai tesque, #04404-84, RRID:AB_10013361, 1:3000), rabbit anti-CGRP (ImmunoStar, #24112, RRID:AB_572217, 1:200), Isolectin GS-IB from Griffonia simplicifolia Alexa Fluor™ 568 Conjugate (2.5 μg/ml, Invitrogen™/Thermo Fisher Scientific, #I21412), Isolectin GS-IB from Griffonia simplicifolia Alexa Fluor™ 647 Conjugate (2.5 μg/ml, Invitrogen™/Thermo Fisher Scientific, #I32450) and rabbit anti-dsRed (1:1000; Takara, RRID:AB_2801258). The following corresponding Alexa Fluor™ conjugated secondary antibodies (1:750; Thermo Fisher Scientific) were used: Alexa Fluor 488 conjugated donkey anti-Rat IgG (Thermo Fisher Scientific, #A48269), Alexa 594 conjugated donkey anti-Rabbit IgG (Thermo Fisher Scientific, #A32754).

## Reverse transcription and quantitative real-time PCR

To compare NGF-induced changes in mRNA expression levels of TMEM100 in different nociceptor subclasses (Fig. 1e), primary L3-4

DRG neurons from WT mice were cultured in the absence and presence of NGF (50 ng/ml) for 24 h before cell collection. To compare mRNA expression levels of TMEM100 in treated (CFA) and control (saline) knee innervating nociceptors at the time of maximum pain (3 dpi), CHRNA3-EGFP+/FB+ and IB4−/FB+ neurons from the ipsi- (CFA) and contralateral (saline) side were collected from acute primary L3-4 DRG cultures of WT mice immediately after adding fresh growth medium without further incubation. In both approaches cultures were counterstained with Alexa Fluor™ −568 conjugated IB4 (2.5 μg/ml, Isolectin GS-IB from Griffonia simplicifolia, Alexa Fluor™ 568 Conjugate, Invitrogen, I21412) for 10–15 min at room temperature to enable the identification of different nociceptor subpopulations.

Samples (20 cells per subpopulation and condition) were manually collected using a fire polished pipette with a tip diameter of ~25 μm pulled (Flaming-Brown puller, Sutter Instruments, Novato, CA, USA) from borosilicate glass capillaries (BF150-86-10, Sutter Instrument) that were filled with 2 μl of picking buffer [1 μL RNAse inhibitor (Takara #2313 A) in 49 μL PBS]. After aspirating 20 cells per sample [NGF ±: CHRNA3+, IB4−, IB4+, Aδ-nociceptors (see Fig. 1e); CFA/Saline: CHRNA3-EGFP+/FB+, IB4−/FB+ (see Fig. 3c)] the pipette was immediately shock frozen in liquid Nitrogen and the cells were expelled into an RNAse free tube filled with 8 μL of picking buffer. Directly thereafter, the tubes were stored at −80 °C until further processing. For each gene 4 to 9 samples (1 sample per subpopulation and condition per mouse) were collected. Cell populations were identified and picked using a 20x magnification and appropriate filter cubes in the Zeiss Axio Observer A1 microscope (Carl Zeiss). Cell lysis and reverse transcription with cDNA synthesis was carried out directly on the sample using the Power SYBR® Green Cells-to-CT™ Kit (Thermo Fischer Scientific, #4402953) following the manufacturer's instructions. qPCR reactions were set up using FastStart Essential DNA Green Master (Roche, #06402712001) according to the manufacturer's guidelines. Per reaction (20 μl reaction volume) 4 μl of the obtained cDNA as template was added to 10 μl SYBR Green PCR Master Mix, 4 μl nuclease-free H2O and the following forward (FW) and reverse (RV) primer pairs (1 μl each of a 5 μM dilution, final concentration: 250 nM): GAPDH-FWD 5′-GCATGGCCTTCCG TGTTC-3′; GAPDH-REV 5′-GTAGCCCAAGATGCCCTTCA-3′; TMEM100-FWD 5′-GAAAAACCCCAAGAGGGAAG-3′; TMEM100-REV 5′-ATGGAAC CATGGGAATTGAA-3′. qPCR reactions were performed in a LightCycler 96 (Roche) with a thermal cycler profile as follows: 10 min pre-incubation step at 95 °C followed by 40 cycles of PCR with a 10 s denaturing cycle at 95 °C, followed by 10 s of annealing at 60 °C and 10 s extension at 72 °C. Mean ± SEM expression levels of TMEM100 normalized to the expression levels of the housekeeping gene GAPDH were compared in the different nociceptor subclasses cultured in absence and presence of NGF (Fig. 1e). CFA-induced changes in mRNA expression levels of TMEM100 in CHRNA3-EGFP+/FB+ and IB4−/FB+ neurons compared to contralateral control neurons were analyzed using the ΔΔCt method (Fig. 3c).

## RNA sequencing

For RNAseq, CHRNA3-EGFP+ samples (20 cells per sample and condition) of 3 WT mice were processed, cultured (±NGF for 24 h) and collected as described above (see Reverse transcription and quantitative real-time PCR). The SmartSeq2 protocol published by Picelli et al.[58] was used to process cell lysates to reverse transcription and library preparation was performed using the Nextera DNA Sample Preparation kit (Illumina) following the manufacturer's instructions. Libraries were sequenced with Illumina HiSeq 2000. Sequencing reads were mapped to GRCm38 mouse reference genome and differential gene expression analysis was performed using the BioJupies platform[59] with default parameters. Next generation RNA-sequencing raw data (FASTQ files) have been deposited in the Gene Expression Omnibus (GEO) under accession number GSE199580 and are publicly available as of the date of publication.

## Calcium Imaging

To examine the responsiveness of FB-labeled neurons to the TRPA1 agonist allylisothiocyanate (AITC, Sigma-Aldrich) Calbryte-590 (Calbryte™ 590 AM, AAT Bioquest) Ca2+-imaging was performed. CHRNA3-EGFP+/FB+ neurons and small (<30 μm) IB4−/FB+ neurons from both WT and TMEM100KO animals after CFA and control treatment were obtained as described above (see Patch clamp recordings), counterstained with Alexa Fluor™ −647 conjugated IB4 (2.5 μg/ml, Isolectin GS-IB from Griffonia simplicifolia, Alexa Fluor™ 647 Conjugate, Invitrogen™/Thermo Fischer Scientific, I32450) for 10–15 min at room temperature, washed with extracellular buffer (140 mM NaCl, 4 mM KCl, 2 mM CaCl2, 1 mM MgCl2, 4 mM glucose, 10 mM HEPES and was adjusted to pH 7.4 with NaOH) and then incubated with the Ca2+ indicator Calbryte-590 (5 μM diluted in ECB from a 5 mM stock solution in DMSO) for 30 min at 37 °C. Coverslips with loaded cells were then washed with ECB, mounted onto a perfusion chamber and superfused with ECB using a constant laminar flow provided through an 8-channel valve controlled gravity-driven perfusion system (VC3-8xG, ALA Scientific Instruments) and a peristaltic pump. A manifold system with 8 inlet ports fitted to a silicon tube bath inlet whose end was positioned at the outer edge of the coverslip without interfering the visual field was used to provide immediate release of ECB and chemical agents into the superfusion chamber. This system enabled minimal dead volume and air bubbles in the lines. Tubes were identical for each input line. All experiments were conducted at room temperature (23 ± 1 °C). Fluorescence images were captured with a Hamamatsu ORCO-Flash4.0 camera at 2 Hz under an inverted Zeiss Axio Observer A1 microscope equipped with a LED light source (CoolLED pE-340fura). ZEN 2 pro software (Carl Zeiss Microscopy GmbH) was employed to detect and analyze intracellular calcium changes throughout the experiment. During imaging the following protocol was applied. After establishing a 30 s baseline with ECB (0–30 s), neurons were challenged with AITC (10 μM) for 60 s (31–90 s) followed by a wash out period of 270 s (91–360 s). At the end of the protocol, 100 mM KCl was applied for 30 s (361–390) to depolarize neurons in order to identify viable neurons in contrast to non-neuronal cells or non-functioning neurons. KCl application was followed by a last wash out period with ECB for 30 s (391–420 s). Neuronal viability was defined as a > 20% increase of fluorescence intensity from the mean intensity 20 s pre-KCl application (330 −350s).

Analysis was conducted by extracting mean intensity values of neurons (CHRNA3-EGFP+/FB+ neurons and IB4−/FB+) after background subtraction from manually drawn regions of interests (ROIs including background ROI) in the ZEN 2 pro software (Carl Zeiss Microscopy GmbH). These values were then transferred into a custom-made Microsoft Excel® template to compute the proportion of neurons responding to AITC under different conditions (CFA/saline; WT/TMEM100KO). In brief, fluorescence is shown as ΔF/F0 with ΔF = F1 − F (F1 = mean intensity of image, F = mean intensity of baseline fluorescence from 0–20 s). Cells responding with an increase >5 % of fluorescence intensity from baseline to AITC application were counted as AITC responders. Cells not crossing the KCl threshold were excluded from the analysis.

## Ex-vivo skin-nerve preparation

To examine peripheral sensitization, we directly measured the mechanosensitivity of C-fiber and Aδ-fiber nociceptors in the tibial nerve by recording mechanically evoked action potentials from single nerve fibers in an ex-vivo skin-nerve preparation. To this end, WT and TMEM100KO mice were sacrificed 3d after CFA/saline injection by placing them in a CO2-filled chamber for 2–4 min followed by cervical dislocation. After dissection, the glabrous skin of the hind limb was placed with the corium side up in a heated (32 °C) organ bath chamber that was perfused with synthetic interstitial fluid (SIF buffer) consisting of 108 mM NaCl, 3.5 mM KCl, 0.7 mM MgSO4, 26 mM NaHCO3, 1.7 mM

Na $H_2PO_4$, 1.5 mM $CaCl_2$, 9.5 mM sodium gluconate, 5.5 mM glucose and 7.5 mM sucrose at a pH of 7.4. The tibial nerve was attached in an adjacent chamber for fiber teasing and single-unit recording. Single units were isolated using a mechanical search stimulus applied with a glass rod and classified by conduction velocity, von Frey hair thresholds and adaptation properties to suprathreshold stimuli[6]. A cylindrical metal rod (diameter 1 mm) that was driven by a nanomotor® (MM2A-LS, 914 Kleindiek Nanotechnik GmbH, Germany) coupled to a force measurement system (FMS-LS, Kleindiek Nanotechnik GmbH, Germany) was used to apply mechanical ramp-and-hold stimuli. The mechanical thresholds of single units were determined by mechanically stimulating the most sensitive spot of the receptive fields using von Frey filaments (Aesthesio® Precision Tactile Sensory Evaluators). The force exerted by the weakest von Frey filament that was sufficient to evoke an action potential was considered as the mechanical threshold. The raw electrophysiological data was amplified with an AC coupled differential amplifier (Neurolog NL104 AC), filtered with a notch filter (Neurolog NL125-6), converted into a digital signal with a PowerLab SP4 (ADInstruments) and recorded at a sampling frequency of 20 kHz using LabChart 7.1 (ADInstruments).

## AAV-PHP.S production

AAV-PHP.S viral particles were produced using a modified protocol based on established procedures by Gradinaru and colleagues[60]. Briefly, AAV-293 cells (Agilent, 240073) were seeded on 150 mm dishes and transfected using polyethylenimine (Polysciences, 23966) with four plasmids: a pAAV of interest (AAV-CAG-dsRedExpress2 or AAV-CAG-TMEM100-IRES-dsRedExpress2, both with AAV2 ITRs), pAdDeltaF6 (Helper, Addgene #112867), pUCmini-iCAP-PHP.S (Addgene #103006) and a mutated pUCmini-iCAP-PHP.S having a 6xHis tag on the VP3 capsid protein[61], with 1:2:2:2 ratio respectively. Cell culture medium was changed at 48 and 120 h post-transfection and supernatant containing viral particles was centrifuged at 1690 g for 10 min. Supernatant medium was filtered (0.2 μm) and diluted in PBS. Cell pellet and filtered medium were stored at 4 °C. At 120 h post transfection, pelleted cells and the ones in the dishes were lysed and incubated at 37 °C for 1 h with the specific PBS buffer containing: $MgCl_2$ 6 mM, Triton X-100 0.4%, RNAse A 6 μg/ml (Roche, #10109169001), DENARASE 250U/μl (c-LEcta, #20804). Lysed cells were collected, diluted in PBS and centrifuged at 2300 g for 10 min. Supernatant from the cell lysate and the filtered medium were incubated separately with equilibrated Ni-sepharose excel histidine-tagged protein purification resin (Cytiva, #17371202) for at least 2 h at room temperature with gentle mixing. Filtered medium and cell lysate were carefully loaded through a gravity flow chromatography column with a 30 μm filter (Econo-pac, Bio Rad, #7321010). Beads were washed with 80 ml of washing buffer (20 mM imidazole in PBS, pH 7.4) and viral particles were then eluted in 50 ml of elution buffer (500 mM imidazole in PBS, pH 7.4). Buffer exchange and concentration was done with Vivaspin 20 ultrafiltration unit having a 1,000,000 molecular weight cut-off (Sartorius). Viral particles were washed and resuspended in PBS and titered using quantitative PCR with primers targeting WPRE element.

## Statistics

Unless otherwise stated, all data are expressed as means ± SEM. All statistical analyses were performed with Microsoft Excel and Prism v8 and v9.0 (Graphpad). Data distribution was systematically evaluated using D'Agostino-Pearson test and parametric or non-parametric tests were chosen accordingly. The statistical tests that were used, the exact P-values and information about the number of independent biological replicates are provided in the display items or the corresponding figure legends. Symbols on graphs (* or #) indicate standard P-value range: *$P < 0.05$; **$P < 0.01$; ***$P < 0.001$ and ns (not significant) $P > 0.05$. Additional information about the statistical tests is provided in the Source Data File.

## Reporting summary

Further information on research design is available in the Nature Portfolio Reporting Summary linked to this article.

## Data availability

All data supporting the findings of this study are available within the article and its supplementary information files. The RNA sequencing dataset generated in this study are deposited in the Gene Expression Omnibus under accession number GSE199580. Source data are provided with this paper. A reporting summary for this article is available as a Supplementary Information file. Source data are provided with this paper.

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

## Acknowledgements

We thank Ms. Anke Niemann and Ms. Claudia Lüchau for technical support. We also acknowledge support from Dr. Claudia Pitzer from the Interdisciplinary Behavioral Core (INBC) at Heidelberg University. This work was funded by the Deutsche Forschungsgemeinschaft grant SFB1158/A01 to S.G.L. and a fellowship from the Physician Scientist Program of the Medical Faculty of Heidelberg University awarded to T.A.N.

## Author contributions

T.A.N., N.W., C.V., P.A., I.S., S.B., J.V., V.P., C.M., N.Z., V.B., A.T.T., C.L.P., F.J.T. and S.G.L. performed the experiments and analyzed data. G.R.L., Y.J.L. and P.A.H. provided material. T.A.N. and S.G.L. designed and supervised the experiments and wrote the manuscript.

## Funding

## Competing interests

The authors declare no competing interests
