## [Peer Review File · Nature Communications]

Role of TMEM100 in mechanically insensitive nociceptor un-silencingREVIEWER COMMENTS

Reviewer #1 (Remarks to the Author):

The present paper builds on a Cell Report published in 2017 showing that some silent nociceptors selectively express CHRNA3. This paper is key to the present work. The sympathetic neuron marker CHRNA3 is found in a small number of mouse DRG neurons – between 15 and 40% of CGRP+ neurons that are themselves considerably less than half the population of nociceptive neurons.

This is a good publishable paper that makes a convincing case for an indirect interaction between TMEM 100 and Piezo2 in a small set of nociceptors that express CHRNA3. Induction of TMEM 100 by the inflammatory mediator NGF causes enhanced Piezo2 activity, though the mechanism of the interaction remains uncertain. Both PKC and NGF have been shown by the Cesare group (1) to enhance mechanotransduction, NGF in a transcriptionally dependent manner, confirmed in the present paper. There are a few issues. First, this set of neurons are only one set of neurons that can be unmasked by various mediators or neuropathic insults. For example, neuropathic injury by things like oxaliplatin (2) unmask a large number of cold (and maybe mechanically-insensitive –this is not clear) insensitive sensory neurons very rapidly. In addition, there is no question that the skin contains silent nociceptors that cannot belong to the class that the authors address (3). The title and text need to be modified to clarify this

Secondly it is not clear how significant the role of the CHRNA3 neurons are in hyperalgesia and allodynia compared to other sets of sensory neurons. The fact that TMEM100 KO mice have normal inflammatory pain suggests not much is going on with these neurons. Are we dealing with a relatively small effect here? Most nociceptors are sensitised by NGF and express TrkA. This issue could be addressed by making a CHRNA3 Cre and crossing it with a PIRT or advillin floxed Diphtheria toxin mouse, and checking the various behavioural models described in the paper. CHRNA3 is massively expressed by sympathetic neurons of course, and there is some low level advillin expression there, so maybe a CreERT2 would be best. I am not at all suggesting that this should be done now – but it would sort the whole area out very nicely. In addition, the use of genetically-encoded calcium indicators would provide useful information about the sets of neurons unmasked by not only NGF but other mediators or insults.

1) Modulation of sensory neuron mechanotransduction by PKC- and nerve growth factor-dependent pathways.

Di Castro A, et al. Proc Natl Acad Sci U S A. 2006 Mar 21;103(12):4699-704.

2) Silent cold-sensing neurons contribute to cold allodynia in neuropathic pain.

MacDonald DI, et al. Brain. 2021 Jul 28;144(6):1711-1726.

3) Slow depolarizing stimuli differentially activate mechanosensitive and silent C nociceptors in human and pig skin.

Rukwied R, et al. . Pain. 2020 Sep 1;161(9):2119-2128.

Reviewer #2 (Remarks to the Author):

This study revealed a new mechanism leading to activation of mechanically insensitive afferents (MIA or silent nociceptors) following inflammation. MIAs are chemoceptors, capable of responding to mediators released from injured or inflamed tissues, whose subsequent sensitization allows them to gain mechanical sensitivity. These investigators previously reported that CHRNA3-EGFP expression marked a large group of unmyelinated MIAs. Here they showed that the expression of the TMEM100 transmembrane protein was induced in cultured CHRNA3-EGFP+ neurons, in response to an inflammatory reagent NGF (nerve growth factor). In vivo intraarticular injection of the inflammatory reagent CFA, which caused the development of arthritis, also led to an increase in TMEM100 expression and a gain of mechanical sensitivity in CHRNA3-EGFP+ neurons. A conditional knockout in nociceptors showed that TMEM100 was necessary for CFA to drive mechanical sensitization in CHRNA3-EGFP+ articular nociceptors. Consequently, TMEM100 knockout led to a loss of CFA-induced secondary sensitization of nociceptors that innervate the hindpaw and caused a loss of sensitized behavioral responses to punctate mechanical stimulation to the paw. Finally, TMEM100 overexpression can sufficiently drive secondary mechanical sensitivity.

Earlier human studies have long indicated the involvement of MIAs in driving tonic pain associated with tissue injury. With their enrichment in deep tissues, MIAs could play crucial roles for driving clinically relevant pain originated from joints and visceral organs. As such, identification of molecules necessary for MIAs to gain mechanical sensitivity is significant. However, a few minor issues need to be addressed or discussed.

To some degree, the lack of impact on primary joint pain in TMEM100 mutants might reduce the significance of the finding. The author should point out that the catwalk assay may not necessarily reveal all spectrum of primary joint pain, and a change in catwalk could also reflect some structural damage in joint, not necessarily a reflection of pain. Ideally, other assays should be considered, if available. Minimally, the authors need to give a careful discussion.

Recent years, people increasingly argued that reflective responses may not measure pain or allodynia (e.g., see PMID: 35016037). All the secondary allodynia should be changed to as “secondary mechanical hypersensitivity”

For the gain-of-function studies, whose interpretation could be complex, the authors do not have to rule out a potential complexity that the induction of secondary mechanical hypersensitivity could also be caused by non-MIA neurons or even by non-neural cells. The significance of the paper is already justified by the strong electrophysiological and behavioral data in mutants.

Reviewer #3 (Remarks to the Author):

Lechner lab has previously discovered a marker *Chrna3*, identifying silent (mechanically insensitive afferents, MIA) nociceptors in mouse. In this study, the authors aim to identify the mechanism by which *Chrna3*⁺ afferents become unsilenced, and how activity in these neurons contributes to joint and skin hyperalgesia in a model of osteoarthritis based on injection of CFA into the knee joint. Using differential gene expression, one gene upregulated in CFA mice is *TMEM100* and they go on to show that *TMEM100* is sufficient and required for unsilencing of MIA, thus turning them from mechano-insensitive to mechano-sensitive. They study *TMEM100* KO mice and find that CFA-induced joint pain is unaffected while skin allodynia fails to develop. *TMEM100* overexpression experiments are also performed and only affects secondary allodynia. The conclusion is that during joint inflammation activity in previously silent joint afferents does not lead to pain, but instead leads to secondary sensitization (for example in spinal cord). Such central sensitization is what triggers cutaneous allodynia, although cutaneous primary afferents (which presumably are *Chrna3*-negative) also become sensitized. Overall, it is a very nice study which advances the concept of silent nociceptors. However, there are a few issues which could be improved.

1. Overall, it was very difficult to follow the actual experiments, because they are not always explained in the result section. Sometimes they are not even described in the materials section and only briefly in figure legend. Thus, despite constantly flipping between result section and materials section to try to understand what was done in each experiment it was still very difficult to review the manuscript. I think the manuscript would benefit by briefly explain the experiments in the result section, rather than just concluding the results. Here are some examples but there are more:

How was qPCR was used to identify expression in different types of neurons (I guess manual picking of cultured IB4⁺ IB4⁻ *Chrna3*⁺ cells? Ad remains unclear even if trying to infer what was done).

How were cells transfected?, in vitro or in vivo?, how was overexpression of *TMEM100* quantified?

For FB traced cells, how was expression compared, change in number of cells or increased in previously expressing cells? Were cells cultured or analyzed in situ?

2. Overall, I think it is essential to quantify overexpression of *TMEM100* in situ in the CFA model and analyze if there is more expression in previously expressing cell types and/or more expressing cells. As far as I can understand from the manuscript, they only did this on cultured cells. Sensory neurons cultured for 24 hours markedly change phenotype and alterations might not represent in vivo changes.

3. Is increased *TMEM100* normalized in vivo (in situ) at a time point of pain resolution?

4. The sensitization experiments in Fig 6 are unclear to me. Sensitization threshold 0.16g are claimed in the text, but mean threshold for activation (first spike) is not analyzed in a ramp. Furthermore, the traces in Fig 6a shows only 0.5 gram as lowest and 25 grams as highest. Furthermore, APs in 6e not different but in 6a, there are clearly more APs in CFA. Thus, there seems to be something not fully aligned in this experiment.

5. In Fig 7, *TMEM* overexpression is not shown, as claimed, instead dsRed is analyzed.

6. I am not sure I understand the AITC/TrpA1 experiment. In Fig. 3, CFA sensitizes neurons to TrpA1 agonist AITC leading to a marked increase of Ca flux while WT neurons hardly respond. In Fig. 5,

TMEM100 KO mice show a marked increase in Ca flux also in saline mice (from basically no response in WT mice of Fig 3 to $\Delta F/F$ 40 in Fig. 5). This increase in TMEM KO mice without any CFA is similar to CFA in WT mice. Thus, removing TMEM100 leads to similar increase in Ca flux as administering CFA in the joint. Next, performing CFA in the TMEM100 KO mice (which already have an increased Ca flux) does not increase it further. This is not their conclusion of this experiment, and the authors might need to summarize their results so that there is a match with data.

7) The idea of central sensitization is based on the assumption that the Cat walk measure joint pain. However, could usage of paws as measured in the Cat walk be affected by cutaneous hyperalgesia? If so, the idea of central sensitization in this manuscript is weak, in particular because they actually have data showing sensitized cutaneous nociceptors while there is no data in the manuscript on central sensitization.

8. I think the results showing that unsilencing silent nociceptors is not causing joint pain is unexpected. This is not really discussed. Does this suggest that other joint afferents are the cause for joint pain? What do the authors believe the function of silent nociceptors in the joint is?

Minor:

Line 33 – unclear what “both channels” are referring to. Can be misunderstood as TrpA1 and TrpV1-

Point-by-point response to the reviewer comments

Reviewer #1 (Remarks to the Author):

The present paper builds on a Cell Report published in 2017 showing that some silent nociceptors selectively express CHRNA3. This paper is key to the present work. The sympathetic neuron marker CHRNA3 is found in a small number of mouse DRG neurons – between 15 and 40% of CGRP+ neurons that are themselves considerably less than half the population of nociceptive neurons.

This is a good publishable paper that makes a convincing case for an indirect interaction between TMEM 100 and Piezo2 in a small set of nociceptors that express CHRNA3. Induction of TMEM 100 by the inflammatory mediator NGF causes enhanced Piezo2 activity, though the mechanism of the interaction remains uncertain. Both PKC and NGF have been shown by the Cesare group (1) to enhance mechanotransduction, NGF in a transcriptionally dependent manner, confirmed in the present paper.

We would like to thank the reviewer very much for the overall positive summary and for considering our manuscript a “good publishable paper”.

Nevertheless, we would like to clarify his/her introductory statement i.e. “...between 15 and 40% of CGRP+ neurons that are themselves considerably less than half the population of nociceptive neurons...”, which is not quite correct. We had previously shown that ~40% of all knee joint and visceral afferents are CHRNA3+ (Prato et al. Cell Reports, 2017) and corroborate these findings in the present study by showing that 47% of all retrogradely labelled nociceptive knee joint afferents are CHRNA3-EGFP+ (Fig. 3b). To the best of our knowledge, no other lab has previously characterized the CHRNA3+ population, yet the reviewer quotes numbers (15-40%) that are different from the numbers we provide here and in our previous study.

With regards to the reviewer’s statement that CGRP+ neurons constitute about half the population of nociceptors, we would like to point out that the distribution of peptidergic (CGRP+) and non-peptidergic nociceptors significantly differs between tissues. Thus, numerous studies have shown that viscera, muscles and joints are almost exclusively innervated by peptidergic CGRP+ afferents, while the skin is innervated by both peptidergic and nonpeptidergic nociceptors, as stated by the reviewer (McMahon et al. Neuron 1994, Bennett et al. Neurosci Lett 1996, Zylka et al. Neuron 2005, Yang et al. Cell Reports 2013, Cranfill & Luo 2021 Curr Top Dev Biol; Ma Quifu Neuron 2022), which underpins the relevance of CGRP+ fibers in general and CHRNA3+ fibers in particular for deep somatic and visceral pain.

Regarding the comment about the findings of the Di Castro et al. paper from the labs of Paolo Cesare and John Wood, which showed that PKC and NGF enhance mechanotransduction, we would like to note that these findings were confirmed by our previous study (Prato et al. Cell Reports 2017), where we also cited and discussed Di Castro et al. in great detail. The present paper, however, focuses on the role of TMEM100 and most importantly on the role of CHRNA3+ silent nociceptors in inflammatory pain and thus has no direct overlap with the study from Di Castro and colleagues. Nevertheless, since the excellent work of the Wood and Cesare labs including the Di Castro et al paper laid a solid foundation for the current study we have now cited Di Castro et al in the revised version of this manuscript.

There are a few issues.

First, this set of neurons are only one set of neurons that can be un silenced by various mediators or neuropathic insults. For example, neuropathic injury by things like oxaliplatin (2) unmask a large number of cold (and maybe mechanically-insensitive –this is not clear) insensitive sensory neurons very rapidly. In addition, there is no question that the skin contains silent nociceptors that cannot belong to the class that the authors address (3). The title and text need to be modified to clarify this

We fully agree that there are other sets of nociceptors that can also be un silenced. The neurons we investigated here become sensitive to mechanical stimuli, whereas the above-mentioned population that is unmasked by the chemotherapeutic drug oxaliplatin becomes cold-sensitive. Therefore, we had already mentioned the oxaliplatin paper in the introduction of the original submission (p3, line 16) and had explicitly pointed out that it is more appropriate to refer to CHRNA3+ afferents as “mechanically insensitive afferents” (MIAs).

To address the reviewer’s concerns, we have – as suggested – revised the manuscript and the title accordingly. Thus, we have replaced the term ‘silent nociceptor’ by MIA throughout the entire manuscript to emphasize that these fibers are only insensitive to mechanical stimuli and we have extended the description of the silent cold-sensing neurons in the introduction (page 3, lines 16-18) to emphasize that there are also other sensory neuron subpopulations that can be un silenced.

We would like to add that to our best knowledge it seems highly unlikely that the silent cold-sensing neurons are also mechanically-insensitive. MacDonald and colleagues did not thoroughly examine mechanosensitivity. They did, however, show that the silent cold-insensitive neurons are a subset of Aδ-fiber nociceptors and it was previously shown that less than 5% of the Aδ-fiber nociceptors in the mouse skin are mechanically-insensitive (Wetzel et al. Nature 2007; PMID: 17167420). Moreover, the transcriptional profile of silent cold-sensing neurons (CGRP+, NF200+, TrkA+, Nav1.8+ and Nav1.6+ but TRPV1-negative, see MacDonald et al. Brain 2021) is highly reminiscent of a subset of mechanosensitive Aδ-nociceptors that are required for the detection of pinprick stimuli, which we have previously described (Arcourt et al. Neuron 2017). Hence, it seems highly unlikely that the silent cold-sensing neurons are also mechanically-insensitive as proposed by the reviewer.

Regarding the reviewers’ statement about the skin also containing silent nociceptors, it is important to note that this is true for human and for pig skin (Rukwied et al. 2020 PMID: 32379219), but there are only very few mechanically

insensitive – i.e. silent – nociceptors in the mouse skin, which was demonstrated using an elegant electrical search protocol by Wetzel and colleagues (Nature 2007; PMID: 17167420).

Secondly it is not clear how significant the role of the CHRNA3 neurons are in hyperalgesia and allodynia compared to other sets of sensory neurons. The fact that TMEM100 KO mice have normal inflammatory pain suggests not much is going on with these neurons. Are we dealing with a relatively small effect here? Most nociceptors are sensitised by NGF and express TrkA.

The observations that TMEM100 KO mice do not develop long-lasting secondary mechanical hypersensitivity and that selective overexpression of TMEM100 in knee joint afferents induces secondary hypersensitivity in the paw, clearly demonstrate that CHRNA3⁺ MIAs significantly contribute to inflammation-induced knee joint pain. We think it is important to clarify that the adjective “secondary” doesn’t implicate that secondary hypersensitivity is less important or painful than primary hypersensitivity, but describes pain hypersensitivity outside the zone of injury or inflammation. In contrast to primary hypersensitivity, which is induced and maintained by peripheral mechanisms and thus usually resolves together with inflammation, secondary hypersensitivity depends on neural plasticity in the spinal cord and often persists long after its initial cause (e.g. inflammation or injury) has resolved and is thought to drive the transition from acute to chronic pain. There is, for example, a strong correlation between the extent of secondary hyperalgesia and the likelihood of developing chronic post-surgical pain (Richebe, P., Anesthesiology 2018). Hence, understanding the mechanisms that induced long-lasting secondary mechanical hypersensitivity, which is the focus of our study, is of major clinical significance.

The statement that “*most nociceptors are sensitized by NGF and express TrkA*” is also not valid. NGF acts via the TrkA receptor, which is only expressed in CGRP⁺ nociceptors and as the reviewer him/herself pointed out above, CGRP⁺ neurons account for app. 50% of all nociceptors (see first comment of the reviewer as well as our reply). By contrast, here he/she claims that most nociceptors are sensitized by NGF, which – with all due respect – is simply not true. A good example showing that NGF doesn’t sensitize all nociceptors is in fact provided in the Di Castro paper that is mentioned by this reviewer, which shows that mechanosensitivity of IB4⁺ neurons (i.e. CGRP-negative neurons) is not altered by NGF.

This issue could be addressed by making a CHRNA3 Cre and crossing it with a PIRT or advillin floxed Diphtheria toxin mouse, and checking the various behavioural models described in the paper. CHRNA3 is massively expressed by sympathetic neurons of course, and there is some low level advillin expression there, so maybe a CreERT2 would be best. I am not at all suggesting that this should be done now – but it would sort the whole area out very nicely.

Thank you very much for this valuable comment. We fully agree that sensory neuron specific ablation of CHRNA3⁺ neurons would be an elegant way to further study the role of these cells in inflammatory pain, but generating a novel transgenic mouse line would be beyond the scope of this revision. We are glad that the reviewer shares our opinion and explicitly wrote that he/she is “...not at all suggesting that this should be done now...”. Thus, we have not made any efforts to generate a Chrna3-CreERT2 as part of the revision, but might consider this approach in future studies.

In addition, the use of genetically-encoded calcium indicators would provide useful information about the sets of neurons unmasked by not only NGF but other mediators or insults.

We agree that information about the modulation of other sets of neurons by other mediators and other insults would also be very interesting. Of course, this is exactly what pain researchers working on sensory afferents focus on – i.e. understanding the contribution of different sensory afferent subclasses to different forms of pain – but examining other sets of neurons as well as other mediators and insults is frankly beyond the scope of a single research paper. *In-vitro*, we had actually already compared the effect of NGF and other inflammatory mediators (e.g. bradykinin, prostaglandin E2, serotonin and histamine) on the mechanosensitivity of CHRNA3⁺ and other nociceptors (Prato et al. Cell Reports 2017) as well as the effect of NGF on voltage-gated sodium channel function in different types of nociceptors (Schaefer et al. Mol Pain 2018). Considering that the reviewer cites MacDonald et al, we do, however, assume that the reviewer refers to *in-vivo* calcium imaging. We thank the reviewer for this valuable comment and interpret this as a general statement pointing us in future directions. We assume that the reviewer does not expect us to solve the plethora of open questions in pain research as part of the revision of this manuscript. We hope the reviewer and the editor agree with us.

1) Modulation of sensory neuron mechanotransduction by PKC- and nerve growth factor-dependent pathways.

Di Castro A, et al. Proc Natl Acad Sci U S A. 2006 Mar 21;103(12):4699-704.

2) Silent cold-sensing neurons contribute to cold allodynia in neuropathic pain.

MacDonald DI, et al. Brain. 2021 Jul 28;144(6):1711-1726.

3) Slow depolarizing stimuli differentially activate mechanosensitive and silent C nociceptors in human and pig skin. Rukwied R, et al. . Pain. 2020 Sep 1;161(9):2119-2128.

Reviewer #2 (Remarks to the Author):

This study revealed a new mechanism leading to activation of mechanically insensitive afferents (MIA or silent nociceptors) following inflammation. MIAs are chemoceptors, capable of responding to mediators released from injured or inflamed tissues, whose subsequent sensitization allows them to gain mechanical sensitivity. These investigators previously reported that CHRNA3-EGFP expression marked a large group of unmyelinated MIAs.

Here they showed that the expression of the TMEM100 transmembrane protein was induced in cultured CHRNA3-EGFP+ neurons, in response to an inflammatory reagent NGF (nerve growth factor). In vivo intraarticular injection of the inflammatory reagent CFA, which caused the development of arthritis, also led to an increase in TMEM100 expression and a gain of mechanical sensitivity in CHRNA3-EGFP+ neurons. A conditional knockout in nociceptors showed that TMEM100 was necessary for CFA to drive mechanical sensitization in CHRNA3-EGFP+ articular nociceptors.

Consequently, TMEM100 knockout led to a loss of CFA-induced secondary sensitization of nociceptors that innervate the hindpaw and caused a loss of sensitized behavioral responses to punctate mechanical stimulation to the paw. Finally, TMEM100 overexpression can sufficiently drive secondary mechanical sensitivity. Earlier human studies have long indicated the involvement of MIAs in driving tonic pain associated with tissue injury. With their enrichment in deep tissues, MIAs could play crucial roles for driving clinically relevant pain originated from joints and visceral organs. As such, identification of molecules necessary for MIAs to gain mechanical sensitivity is significant. However, a few minor issues need to be addressed or discussed.

We thank the reviewer for the nice summary of our work and for emphasizing the significance of our findings.

To some degree, the lack of impact on primary joint pain in TMEM100 mutants might reduce the significance of the finding.

As already pointed out in our response to reviewer #1, we believe that it is a common misconception that secondary hypersensitivity is less problematic or important than primary hypersensitivity, because primary hypersensitivity usually resolves together with inflammation whereas secondary hypersensitivity is thought to drive the transition from acute to chronic pain. Hence, the lack of impact on primary joint pain does not reduce the significance of the finding. We hope this answer together with the more detailed reply to the same comment of reviewer #1 resolves this issue.

The author should point out that the catwalk assay may not necessarily reveal all spectrum of primary joint pain, and a change in catwalk could also reflect some structural damage in joint, not necessarily a reflection of pain.

Ideally, other assays should be considered, if available. Minimally, the authors need to give a careful discussion.

We fully agree with the reviewer and have thus added a careful discussion of the limitations of the catwalk assay to the revised version of the paper. Moreover, as suggested by the reviewer, we have performed additional assays to obtain an alternative measure of CFA-induced pain. Thus, we have used the LABORAS system (laboratory animal behavior observation registration and analysis system, METRIS b.v.) to detect and compare behaviors that might be indicative of pain (e.g. grooming, drinking, immobility, rearing etc.) during observation periods of 16 hours before and after CFA treatment. Interestingly, we did not observe any significant differences between saline and CFA-treated mice before and after treatment in WT mice in behaviors other than rearing and distance walked, which are similar to the parameter assessed with the Catwalk XT device (page 6, lines 15-23). Hence, we did not make efforts to also perform these experiments in TMEM100 KO mice. We did, however, utilize the LABORAS system to analyze possible changes in pain-behaviors after AAV-mediated over-expression of TMEM100 in articular afferents to complement the results shown in the original Fig. 7c and d. Again, no change in the pain-related behaviors detected with the LABORAS system were observed. We would like to note that the intra-articular CFA injection induces a mild and transient form of mono arthritis, which obviously induces localized pain but doesn't alter the overall well-being of the animals enough to cause detectable changes in other behaviors. We have added the results from the LABORAS assays to the supplementary information (Supplementary Fig. 2 and 5) of the revised manuscript and hope that this resolves this issue.

Recent years, people increasingly argued that reflective responses may not measure pain or allodynia (e.g., see PMID: 35016037). All the secondary allodynia should be changed to as "secondary mechanical hypersensitivity"

As suggested we have replaced 'secondary allodynia' with "secondary mechanical hypersensitivity" throughout the manuscript, which resolves this issue (see also page 6, lines 31-33 and page 7, lines 1-2).

For the gain-of-function studies, whose interpretation could be complex, the authors do not have to rule out a potential complexity that the induction of secondary mechanical hypersensitivity could also be caused by non-MIA neurons or even by non-neural cells. The significance of the paper is already justified by the strong electrophysiological and behavioral data in mutants.

We thank the reviewer for pointing out the significance of our electrophysiological and behavioral data and have – as suggested – shortened the section of the discussion where we rule out the contribution of other factors.

Reviewer #3 (Remarks to the Author):

Lechner lab has previously discovered a marker *Chrna3*, identifying silent (mechanically insensitive afferents, MIA) nociceptors in mouse. In this study, the authors aim to identify the mechanism by which *Chrna3*+ afferents become silenced, and how activity in these neurons contributes to joint and skin hyperalgesia in a model of osteoarthritis based on injection of CFA into the knee joint. Using differential gene expression, one gene upregulated in CFA mice is TMEM100 and they go on to show that TMEM100 is sufficient and required for unsilencing of MIA, thus turning them from mechano-insensitive to mechano-sensitive. They study TMEM100 KO mice and find that CFA-induced joint pain is unaffected while skin allodynia fails to develop. TMEM100

overexpression experiments are also performed and only affects secondary allodynia. The conclusion is that during joint inflammation activity in previously silent joint afferents does not lead to pain, but instead leads to secondary sensitization (for example in spinal cord).

Such central sensitization is what triggers cutaneous allodynia, although cutaneous primary afferents (which presumably are Chrna3-negative) also become sensitized. Overall, it is a very nice study which advances the concept of silent nociceptors. However, there are a few issues which could be improved.

We thank the reviewer for his/her positive summary underlining that our results advance the concept of silent nociceptors and his/her constructive input.

1. Overall, it was very difficult to follow the actual experiments, because they are not always explained in the result section. Sometimes they are not even described in the materials section and only briefly in figure legend. Thus, despite constantly flipping between result section and materials section to try to understand what was done in each experiment it was still very difficult to review the manuscript. I think the manuscript would benefit by briefly explain the experiments in the result section, rather than just concluding the results. Here are some examples but there are more:

As suggested we have rewritten the results section accordingly to improve intelligibility.

How was qPCR was used to identify expression in different types of neurons (I guess manual picking of cultured IB4+ IB4- Chrna3+ cells? Ad remains unclear even if trying to infer what was done).

The qPCR and specifically the procedure of sample collection was described in great detail in the method section (subsection "Reverse transcription and quantitative real-time PCR") where we wrote: "Samples (20 cells per subpopulation and condition) were manually collected using a fire polished pipette with a tip diameter of ~25 μm pulled (Flaming-Brown puller, Sutter Instruments, Novato, CA, USA) from borosilicate glass capillaries (BF150-86-10, Sutter Instrument) that were filled with 2μl of picking buffer [1μL RNase inhibitor (Takara #2313A) in 49 μL PBS]. After aspirating 20 cells per sample [NGF ±: CHRNA3+, IB4-, IB4+, Aδ-nociceptors (see Fig. 1e); CFA/Saline: CHRNA3-EGFP+/FB+, IB4-/FB+ (see Fig. 3c)] the pipette was immediately shock frozen in liquid Nitrogen and the cells were expelled into an RNase free tube filled with 8 μL of picking buffer." Moreover, the mouse lines that were used to identify CHRNA3+ silent nociceptors and Npy2r+ A-delta nociceptors are described in the legend of figure 1 ("Tg(Npy2r-cre)SM19Gsat/Mmucd x B6;129S-Gt(ROSA)26Sortm32(CAG-COP4*H134R/EYFP)Hze/J (Npy2rCre;Chr2-EYFP)"). We believe this should be sufficient detail to understand – as the reviewer correctly guessed – that the cells were manually picked as this is explicitly mentioned.

However, to further improve intelligibility, we revised the result section such that it now provides additional details about the experimental procedure.

How were cells transfected?, in vitro or in vivo?, how was overexpression of TMEM100 quantified?

We assume the reviewer refers to the data presented in figure 1f-h. We have mentioned in the method section that the DRG neurons were transfected in-vitro where we wrote "For the experiments shown in Fig. 1f – h, primary DRG cultures were transfected using the Amaxa Nucleofector 4D (Lonza) following the manufactures instructions", but we agree that we should have also indicated this more clearly in the results section. Unfortunately, the company Lonza – as most other companies these days – does not provide any information about the composition of their transfection buffers or about the electrical pulse pattern that is used to drive DNA into the cells. Hence, we could only state that we followed the manufacturer's instructions. To improve intelligibility, we added a brief explanation of the transfection to the result section (Page 5, Lines 6-7)

For FB traced cells, how was expression compared, change in number of cells or increased in previously expressing cells? Were cells cultured or analyzed in situ?

2. Overall, I think it is essential to quantify overexpression of TMEM100 in situ in the CFA model and analyze if there is more expression in previously expressing cell types and/or more expressing cells. As far as I can understand from the manuscript, they only did this on cultured cells. Sensory neurons cultured for 24 hours markedly change phenotype and alterations might not represent in vivo changes.

The TMEM100 expression levels during CFA-induced inflammation were quantified by qPCR from FB-traced cells manually picked from acutely dissociated (2-3 hours) DRG cultures (see Fig. 3c) – **NOT** 24 hours in culture as stated by the reviewer in order to avoid the changes mentioned by the reviewer. Collecting cells from acutely dissociated DRGs is the standard approach that is also used by most recently published large scale single cell RNAseq screens. Importantly, we compared the expression levels of contralateral and ipsilateral (CFA side) neurons and found upregulation of TMEM100 only in ipsilateral neurons, which demonstrates that the upregulation of TMEM100 is triggered by the CFA-induced inflammation and not by the culturing process per se.

Nevertheless, we agree that an in-situ analysis – we assume the reviewer thought about immunohistochemistry – would also be interesting and have thus purchased three different anti-TMEM100 antibodies to address the reviewers concern. Unfortunately, none of the three antibodies specifically recognized TMEM100 as all three also gave strong signals in DRG sections from TMEM100-KO mice (see Figure I below). Hence, we cannot provide an in-situ quantification of TMEM100 overexpression after CFA as suggested by the reviewer. However, for the above-mentioned reasons, we believe that the qPCR data is sufficient to convincingly demonstrate the TMEM100 is indeed up-regulated in CHRNA3+ neurons after CFA treatment. We hope the reviewer as well as the editor agree.

Figure I. Evaluation of the indicated anti-TMEM100 antibodies in DRG sections from WT and TMEM100-KO mice

3. Is increased TMEM100 normalized in vivo (in situ) at a time point of pain resolution?

This is an excellent point. We had not looked at later time points in the original submission, but have now performed additional qPCR analysis of TMEM100 expression in retrogradely labelled knee joint afferents at the end of the observation period (21 days post CFA) and have added this data to figure 3c.

4. The sensitization experiments in Fig 6 are unclear to me. Sensitization threshold 0.16g are claimed in the text, but mean threshold for activation (first spike) is not analyzed in a ramp.

We would like to clarify that we did not claim a “sensitization threshold” of 0.16 g in the text as stated by the reviewer. What we wrote was that “40% of the cutaneous C-fiber nociceptors from mice with CFA-induced knee monoarthritis, are activated by forces of 0.16 g and below and virtually all C-fibers (90%) responded to von Frey filaments of 0.4 g and below (Fig. 6a–d).”

The reason for showing the cumulative threshold distribution of C-fiber nociceptors determined with von Frey filaments (Fig. 6d) was to enable the direct comparison of skin-nerve data with paw withdrawal thresholds from the same mice (Fig. 4b, right panel), which were determined with the very same von Frey filaments. Forces measured with the force sensor attached to the piezo-driven mechanostimulator that is used to apply ramp-and-hold stimuli (Fig. 6a and e) – which is what the reviewer would like to see – cannot be directly compared with von Frey thresholds, because the metal rod that is used to indent the skin with the mechanostimulator has a diameter of 1 mm for all stimuli, whereas the diameter of the von Frey filaments is much smaller and increases with strength (Figure IIA). Accordingly, the pressure (=force/area) produced by a 0.6 g von Frey filament equals $\sim 180 \text{ mN/mm}^2$, whereas the pressure exerted by 0.6 g ($0.6 \text{ g} \times 9.81 \text{ m/s} = 5.9 \text{ mN}$) applied with the mechanostimulator via a metal rod with a diameter of 1 mm (contact area $0.5^2 \times \pi = 0.78 \text{ mm}^2$) equals $\sim 7.5 \text{ mN/mm}^2$. Hence, identical forces exert significantly different pressures depending on which instrument they are applied with. Directly comparing pressures is, however, also not appropriate because due to the relatively large receptive field size of C-fiber nociceptors, their activation thresholds also vary with the size (contact area) of the stimulator.

I hope this explanation clarifies why it is important to determine the von Frey thresholds of C-fibers in skin-nerve recordings and why we chose to show the cumulative distribution of these thresholds.

Should the reviewer or the editor, nevertheless, insist that we show the mean thresholds measured in ramps or the mean von Frey thresholds, we are of course happy to replace the original figure 6d with one of the graphs shown below (Fig. IIB, mean von Frey thresholds or Fig. IIC mean thresholds measured from ramps).

Figure II, A, photo of three von Frey filaments (0.07g, 0.16g, 0.6g) used to measure paw withdrawal thresholds and mechanical activation thresholds in skin-nerve recording and of the metal rod that is mounted to the mechanical stimulator used to apply ramp-and-hold stimuli. **B,** mean mechanical activation thresholds of C-fiber nociceptors determined with von Frey filaments. **C,** mean mechanical activation thresholds of C-fiber nociceptors determined by measuring the force during at which the first

action potential is triggered during the mechanical ramp-and-hold stimuli shown in the original figure 6a and e.

Furthermore, the traces in Fig 6a shows only 0.5 gram as lowest and 25 grams as highest. Furthermore, APs in 6e not different but in 6a, there are clearly more APs in CFA. Thus, there seems to be something not fully aligned in this experiment.

As with all biological data, there is also variability in the firing rate of C-fiber nociceptors in skin-nerve recordings and the traces we had shown in the original submission were examples from the upper and lower end of the spectrum. We agree that those example traces were probably not the most representative ones and have thus replaced them with more appropriate traces in the revised manuscript. The reason for showing only responses evoked by 0.5 to 25 grams was the space limitation in the figure (spike trains are not resolved properly when the images are compressed horizontally).

We hope this explanation resolves this issue.

5. In Fig 7, TMEM overexpression is not shown, as claimed, instead dsRed is analyzed.

TMEM100 overexpression was mediated by an AAV-PHP.S virus in which the TMEM100 sequence is coupled to dsRed via an IRES (internal ribosomal entry site) cassette, with TMEM100 being at the 5-prime end and dsRed being downstream of the IRES cassette. Using the IRES cassette to co-express fluorescent reporter genes is a commonly used and well-established standard technique. Hence, dsRed can only be expressed when TMEM100 is also transcribed and therefore we think it is appropriate to conclude that TMEM100 is expressed in dsRed-positive neurons (page 11, lines 24-25).

6. I am not sure I understand the AITC/TrpA1 experiment. In Fig. 3, CFA sensitizes neurons to TrpA1 agonist AITC leading to a marked increase of Ca flux while WT neurons hardly respond. In Fig. 5, TMEM100 KO mice show a marked increase in Ca flux also in saline mice (from basically no response in WT mice of Fig 3 to deltaF/F 40 in Fig. 5). This increase in TMEM KO mice without any CFA is similar to CFA in WT mice. Thus, removing TMEM100 leads to similar increase in Ca flux as administering CFA in the joint. Next, performing CFA in the TMEM100 KO mice (which already have an increased Ca flux) does not increase it further. This is not their conclusion of this experiment, and the authors might need to summarize their results so that there is a match with data.

The reviewer is right in that saline treated TMEM KO mice already exhibit bigger AITC-induced responses and we sincerely apologize for not having properly explained the results of the Ca²⁺-imaging experiments in detail.

The point that we actually wanted to make was that the proportion of CHRNA3⁺ neurons that respond to AITC is significantly increased after CFA treatment in WT but not in KO mice, which was shown in the bar graphs in Fig. 3h and 5e. For the reviewer's convenience and to allow the direct comparison of the proportions of AITC-sensitive cell from all four conditions, we have created a single bar graph (see figure III below), which shows that the proportion of responders is significantly higher in CFA-treated WT mice compared to all other conditions and genotypes. Moreover, the proportions of AITC responders in CFA-treated KO mice, saline-treated KO mice and saline-treated WT mice do not differ from one another (proportions were compared with Fisher's exact test and corresponding P-values are shown in the figure below). We believe that the increase in the proportion of AITC-sensitive neurons (8% responding in saline mice compared to 41% responding in CFA mice) has a much stronger impact on pain sensitivity than the subtle increase in the response magnitude of individual cells and we have revised the result section and the discussion of the paper accordingly.

Figure III, comparison of the proportions of CHRNA3⁺ knee joint afferents from the indicated mice that respond to the TRPA1 agonist AITC. Comparisons were made with Fisher's exact test and the corresponding P-values are provided above the bars.

We have nevertheless considered the reviewers concern regarding the magnitude of the responses in TMEM100-KO mice and have thus repeated the Ca²⁺-imaging experiments in saline- and CFA-treated KO mice. These experiments (see figure below) corroborated our original results and showed that only very few CHRNA3⁺ articular afferents respond to AITC in saline-treated WT mice and that there is no significant change in the proportion after intraarticular CFA injection. As previously, the response magnitude of CHRNA3⁺ neurons in saline treated KO mice was relatively big and did not significantly change after CFA treatment. In fact, in the new dataset the average response magnitude was even smaller in CFA-treated mice. It is, however, important to note that in the new dataset

the response magnitude depends on two and three responding cells in the saline and CFA-treated mice, respectively, and is thus not very meaningful, because there is considerable variability in the response magnitude as evident from the recordings of the other nociceptor populations that were made in the same cultures (IB4-/FB+, Fig. IVE-H and IB4+/FB-, Fig. IVI-L). The N-number was, in fact, also small in our previous experiments in the saline treated WT mice, where only 4 from 50 CHRNA3+ neurons responded to AITC (see original Fig. 3h), which could explain why the average response was much smaller than the average response of CHRNA3 WT-CFA cells and CHRNA3 KO-saline and KO-CFA cells. We have thus toned down the interpretation of the changes in the response magnitude and have emphasized the change in proportion (see e.g. page 8, lines 6-15), which is more appropriate, in the revised version of the manuscript. Moreover, we have noticed a mistake in the original analysis where the $\Delta F/F_0$ values had been multiplied with the factor 100 and have corrected this in the revised version. We hope that this resolves the issue.

TMEM100-KO

Figure IV. Ca^{2+} -imaging of AITC responses from CHRNA3+ neurons retrogradely labeled with FastBlue (CHRNA3+/FB+, A–D), IB4-/FB+ neurons (peptidergic nociceptors, E–H) and IB4+ but FB- neurons from the same cultures (I–L). Panels A, B, E, F, I and J show the responses of individual cells (non-responders in light grey). Panels C, G and K show the averages of the responders shown in A, B, E, F, I and J. The bar graphs in D, H and L show proportion of cells that respond to AITC in the indicated conditions.

7) The idea of central sensitization is based on the assumption that the Cat walk measure joint pain. However, could usage of paws as measured in the Cat walk be affected by cutaneous hyperalgesia? If so, the idea of central sensitization in this manuscript is weak, in particular because they actually have data showing sensitized cutaneous nociceptors while there is no data in the manuscript on central sensitization.

The Catwalk XT system is the gold standard to measure knee joint pain and outcome measures are not influenced by cutaneous hyperalgesia. This is nicely demonstrated by two observations: Firstly, the stride parameters measured with the Catwalk already exhibit almost maximum alteration 1 day after CFA injection (Fig. 4a), whereas the cutaneous mechanical hypersensitivity in the paw peaks only after 3 days. If the Catwalk would indirectly assess cutaneous hypersensitivity, then the time courses should be identical, which is, however, not the case. Secondly,

the in-vivo TMEM100 overexpression experiments shown in figure 7, clearly showed that mice that have severe mechanical hypersensitivity in the paw (Figure 7d) do NOT exhibit altered gait in the Catwalk (Fig 7c). We believe that these two observations convincingly demonstrate the hypersensitivity in the paws does not affect the outcome of the catwalk assay.

Regarding the reviewers comment about central sensitization, we would like to note that the idea of central sensitization is NOT based on the assumption that the Catwalk measures joint pain. In fact, we do not claim that we have shown that central sensitization contributes to the development of secondary mechanical hypersensitivity, but repeatedly state that it was previously proposed by OTHERS that central sensitization contributes to secondary mechanical hypersensitivity.

Here are a few examples from our original submission:

- Page 9, line 15: *“It is well established that secondary mechanical allodynia results from central sensitization – i.e. a strengthening of synaptic transmission in pain processing circuits in the spinal cord.”*
- Page 10, line 9: *“Hence, our data suggests that in addition to central sensitization, sensitization of cutaneous C-fiber and A δ -fiber nociceptors might also significantly contribute to secondary mechanical allodynia in the hind paw induced by knee joint inflammation.”*
- Page 11, line 23: *“While we have not examined the well-established contribution of central sensitization, our data reveals a previously unrecognized contribution of peripheral sensitization to secondary allodynia.”*
- Page 13, line 15: *“While the spinal mechanisms of central sensitization have been studied extensively in various rodent pain models, including experimentally induced arthritis, ...”*
- Page 14, line 17: *“We have not explicitly tested if un-silencing of MIAs also triggers central sensitization, which is thought to be the major cause of secondary hypersensitivity.”*
- Page 14, line 21: *“...it seems highly likely that central processing of nociceptive input was also altered in these experiments such that subliminal sensory inputs evoked pain.”*

We believe that these statements clearly show that the focus of our study was not on central sensitization.

In fact, the key message of our study was that we have identified TMEM100 as a key regulator of silent nociceptor un-silencing and that our work revealed the hitherto unrecognized physiological role of silent nociceptors in triggering spatially remote secondary allodynia during inflammation.

What we do claim is that the sensitization of nociceptors that innervate skin regions remote from the site of inflammation (knee joint) is induced by the un-silencing of silent nociceptors in the knee joint and we propose that this sensitization is triggered by some sort of central crosstalk between articular silent nociceptors and cutaneous nociceptors (see schematic in Fig. 7h).

Since this message did not seem to come across properly – considering the reviewers comment – we have revised the discussion accordingly to clearly distinguish between central sensitization and central crosstalk. We have also carefully reconsidered the use of the term “peripheral sensitization”, because this term is preoccupied and interpreted as a local sensitization of nociceptors by inflammatory mediators. This is, however, not what we observed. Hence, to avoid confusion and misunderstandings, we now distinguish between peripheral sensitization and “centrally-driven sensitization of cutaneous nociceptors”.

We hope this revision has improved intelligibility and conveys the take-home message better than the original version and that this resolves this issue.

8. I think the results showing that un-silencing silent nociceptors is not causing joint pain is unexpected. This is not really discussed. Does this suggest that other joint afferents are the cause for joint pain? What do the authors believe the function of silent nociceptors in the joint is?

The key point of the paper is that articular silent nociceptors do NOT cause primary joint pain but instead trigger secondary mechanical hypersensitivity in remote skin regions. This observation together with the finding that TMEM100 up-regulation mediates the un-silencing of silent nociceptors is the take home message and the major conceptual advance of the paper, because hitherto it was completely unclear which afferent subtype triggers secondary hypersensitivity.

We thought that we had made this point very clear in the introduction (page 2, line 12ff), the discussion (Page 14, line 24ff) and not least by the cartoon shown in the Fig. 7h. However, since the reviewer had issues with this point, our explanations were probably not clear enough and we have thus edited the manuscript accordingly in order to emphasize this point. We hope the reviewer finds the revised version more intelligible.

Minor:

Line 33 – unclear what “both channels” are referring to. Can be misunderstood as TrpA1 and TrpV1- Thank you very much for the thorough revision. It should in fact be understood as TRPA1 and TRPV1. The sentence has been rephrased to avoid misunderstandings.

REVIEWERS' COMMENTS

Reviewer #1 (Remarks to the Author):

A splendid response to my concerns. The use of the Rs-lkn methodology always pays dividends in terms of publications.

Reviewer #2 (Remarks to the Author):

The authors have addressed my comments. Identification of molecular mechanisms sensitizing silent nociceptors is important for the pain field. Two minor comments without involvement of new experiments:

Since authors acknowledge that the sample sizes are too small and lack sufficient power to conclude the amplitude of AITC-evoked Ca^{++} responses, this set of data could be deleted; just stating a change in proportion of responsive cells could be sufficient for this study (otherwise, an increase of sample sizes are needed to consolidate this conclusion).

Regarding secondary hypersensitivity, the authors considered some forms of crosstalk involved with spinal cord; the authors might also consider another possibility that inflammation in the joint might sensitize the nerve fibers passing through the joint regions and terminating in the hind paw.

Reviewer #3 (Remarks to the Author):

The authors have addressed my concerns with satisfaction.

Point-by-point response to the reviewer comments

Reviewer #1 (Remarks to the Author):

A splendid response to my concerns. The use of the Rs-lkn methodology always pays dividends in terms of publications.

We are happy to hear that our revision and response satisfy the reviewer.

Reviewer #2 (Remarks to the Author):

The authors have addressed my comments. Identification of molecular mechanisms sensitizing silent nociceptors is important for the pain field. Two minor comments without involvement of new experiments:

We are happy to hear that our revision and response satisfy the reviewer.

Since authors acknowledge that the sample sizes are too small and lack sufficient power to conclude the amplitude of AITC-evoked Ca^{++} responses, this set of data could be deleted; just stating a change in proportion of responsive cells could be sufficient for this study (otherwise, an increase of sample sizes are needed to consolidate this conclusion).

We prefer not to delete the data and thus shifted it to the supplementary information as a new Supplementary Figure 3. We believe that it is important to show the time course of the response for reasons of scientific transparency and to show how the experiments were performed. To ensure that the data is not misinterpreted by the readers, we have emphasized the small N-number in the text and note that definitive conclusions about possible differences in the response magnitude cannot be drawn (page 8, line 12-15 and page 9, line 30-32). We hope this is acceptable. If not, we are of course happy to completely delete the data from the manuscript.

Regarding secondary hypersensitivity, the authors considered some forms of crosstalk involved with spinal cord; the authors might also consider another possibility that inflammation in the joint might sensitize the nerve fibers passing through the joint regions and terminating in the hind paw.

This is a good point and in fact one of the reasons why we performed the knee joint-specific TMEM100 overexpression experiment in the first place – i.e. to rule out the spreading of inflammation or – as suggested by the reviewer – inflammation-induced sensitization of nerve fibers passing through the joint region contribute to secondary hypersensitivity. Since selective overexpression of TMEM100 in knee-joint afferents in the absence of an inflammation, however, also causes secondary mechanical hypersensitivity, we believe that this possibility can be ruled out. We have added a few lines to the discussion (page 14 line 34 – page 15 line 2) to explain this.

Reviewer #3 (Remarks to the Author):

The authors have addressed my concerns with satisfaction.

We are happy to hear that our revision and response satisfy the reviewer.